# Nonlinear feedback drives homeostatic plasticity in H$_2$O$_2$ stress response

**Youlian Goulev**[1,2,3,4]*, **Sandrine Morlot**[1,2,3,4], **Audrey Matifas**[1,2,3,4], **Bo Huang**[5], **Mikael Molin**[6], **Michel B Toledano**[5], **Gilles Charvin**[1,2,3,4]*

[1]Developmental Biology and Stem Cells Department, Institut de Génétique et de Biologie Moléculaire et Cellulaire, Strasbourg, France; [2]Centre National de la Recherche Scientifique, Illkirch, France; [3]Institut National de la Santé et de la Recherche Médicale, Illkirch, France; [4]Université de Strasbourg, Illkirch, France; [5]Oxidative Stress and Cancer, IBITECS, SBIGEM, CEA-Saclay, Gif-sur-Yvette, France; [6]Department of Chemistry and Molecular Biology, University of Gothenburg, Gothenburg, Sweden

**Abstract** Homeostatic systems that rely on genetic regulatory networks are intrinsically limited by the transcriptional response time, which may restrict a cell's ability to adapt to unanticipated environmental challenges. To bypass this limitation, cells have evolved mechanisms whereby exposure to mild stress increases their resistance to subsequent threats. However, the mechanisms responsible for such *adaptive homeostasis* remain largely unknown. Here, we used live-cell imaging and microfluidics to investigate the adaptive response of budding yeast to temporally controlled H$_2$O$_2$ stress patterns. We demonstrate that acquisition of tolerance is a systems-level property resulting from nonlinearity of H$_2$O$_2$ scavenging by peroxiredoxins and our study reveals that this regulatory scheme induces a striking hormetic effect of extracellular H$_2$O$_2$ stress on replicative longevity. Our study thus provides a novel quantitative framework bridging the molecular architecture of a cellular homeostatic system to the emergence of nonintuitive adaptive properties.

*For correspondence: youlian.
goulev@gmail.com (YG); charvin@
igbmc.fr (GC)

**Competing interests:** The authors declare that no competing interests exist.

## Introduction

Homeostatic systems are ubiquitous in biology and function to restore internal physiological variables to a given set point following fluctuations in the internal or external environment. The accuracy of such control mechanisms (i.e. its ability to reach an equilibrium state that is as close as possible to the pre-existing state), is thought to be essential to ensure robust physiological adaptation. Therefore, understanding the mechanisms underlying accurate control in regulatory networks has emerged as a central question in Control Theory applied to biological systems (*Khammash, 2016*; *Ma et al., 2009*; *Whitacre, 2012*). Several seminal studies have pointed to the existence of 'perfectly adapting' systems in which exact restoration of the pre-existing state is observed, including bacterial chemotaxis (*Barkai and Leibler, 1997*; *Berg and Tedesco, 1975*), calcium signaling (*El-Samad et al., 2002*), and yeast hyperosmolarity response (*Muzzey et al., 2009*), all of which are based on a regulatory scheme referred to as 'integral feedback' (*Yi et al., 2000*).

However, high control accuracy alone is insufficient to protect against the potentially damaging effects of environmental challenges, and other dynamical properties of homeostatic systems may determine the cell's ability to adapt: indeed, most stress regulatory pathways feature transcriptional responses, which are intrinsically slower processes than the other biochemical effects of stress exposure (*Muzzey and van Oudenaarden, 2009*; *Young et al., 2013*). Because of this limiting response time, it is expected that a transient peak in the internal stress level (output peak of magnitude O$_{max}$, *Figure 1A*) may occur in response to stepwise exposure to an external stressor (input, *Figure 1A*).

**eLife digest** Harmful external conditions, such as extreme heat or radiation, can cause stress to cells that may lead to permanent damage and even death. Cell stress is responsible for some cancers and degenerative diseases, and is involved in the process of aging. Cells respond to stress by modifying their activities in order to prevent damage from occurring. Some studies have suggested that the ability of cells to survive a stressful situation might depend both on the severity of the stress and also on the way in which the stress is applied. For example, the stress might start suddenly or develop more gradually.

Cells exposed to a mild level of stress develop a tolerance that enables them to survive stronger doses of the same stress in the future. However, it is not clear how cells acquire such tolerance, and whether mild levels of stress can have more general benefits to cells, such as increased lifespan.

Hydrogen peroxide and other "oxidative" compounds play important roles in cells, but they are also capable of causing damage so their levels must be tightly controlled. Goulev et al. developed a "microfluidic" device to study the effects of oxidative stress on yeast cells. The device made it possible to precisely control the level of hydrogen peroxide in the cells' environment while monitoring the cells' stress responses.

The experiments show that exposing yeast cells to gradually increasing levels of hydrogen peroxide can train the cells to be able to survive when they are exposed to high levels of this compound. This ability depends on the activity of specific enzymes called peroxidases that are known to be able to destroy hydrogen peroxide inside the cells. The experiments suggest that gradually increasing levels of hydrogen peroxide trigger increases in the production of peroxidases that protect the cells against future oxidative stress.

Further experiments show that even a very low dose of hydrogen peroxide is sufficient to activate the production of the enzymes, leading to an increase in the lifespan of the cells. A future challenge will be to investigate whether the principles identified in this work also apply to other stress responses in yeast.

Such a transient overshoot may trigger irreversible deleterious effects leading to cell death, irrespective of the ability of the homeostatic system to accurately restore the pre-existing steady state ($O_{eq}$ on *Figure 1A*). In this case, interestingly, the rate at which stress is applied (while keeping constant the overall magnitude of stress) should directly control $O_{max}$ and hence determine cellular stress resistance (*Figure 1A*). Remarkably, this hypothesis has neither been formally addressed theoretically nor been tested experimentally. Yet, whether the adaptation range of a homeostatic system is set only by the overall stressor level, or alternatively, depends on the kinetics of the input stress pattern remains a question of fundamental importance.

Interestingly, however, there is one prominent example of stress pattern that confers improved cellular adaptation despite fast environmental changes, known as acquired stress resistance, or stress tolerance: in this case, a mild stress preconditioning increases resistance to subsequent, acute exposure to large doses of the same stressor. This effect has been observed in a broad spectrum of species, from unicellular organisms to mammals, in response to diverse environmental challenges (*Davies et al., 1995*; *Hecker et al., 2007*; *Kandror et al., 2004*; *Kensler et al., 2007*; *Lewis et al., 1995*; *Lindquist, 1986*; *Lou and Yousef, 1997*; *Lu et al., 1993*; *Scholz et al., 2005*) and is considered to be an anticipation strategy to overcome potentially harmful environmental conditions in the future (*Mitchell et al., 2009*). Stress resistance is thus itself an adaptive trait reflecting an intrinsic plasticity of the homeostatic machinery. However, the mechanisms underlying such *adaptive homeostasis*- the robustness of which is improved following stress exposure (*Davies, 2016*), remain to be elucidated. In particular, it is not known how this acquired stress resistance bypasses/overcomes the intrinsically slow response time of the stress response (e.g. through faster transcriptional response, or initially higher stressor degradation rate- analogous to stress buffering) and how this might be mechanistically achieved (*Figure 1A*).

Here, we used the response of budding yeast to hydrogen peroxide ($H_2O_2$) stress as a model system to investigate the driving principles that govern the adaptation to arbitrary stress patterns and

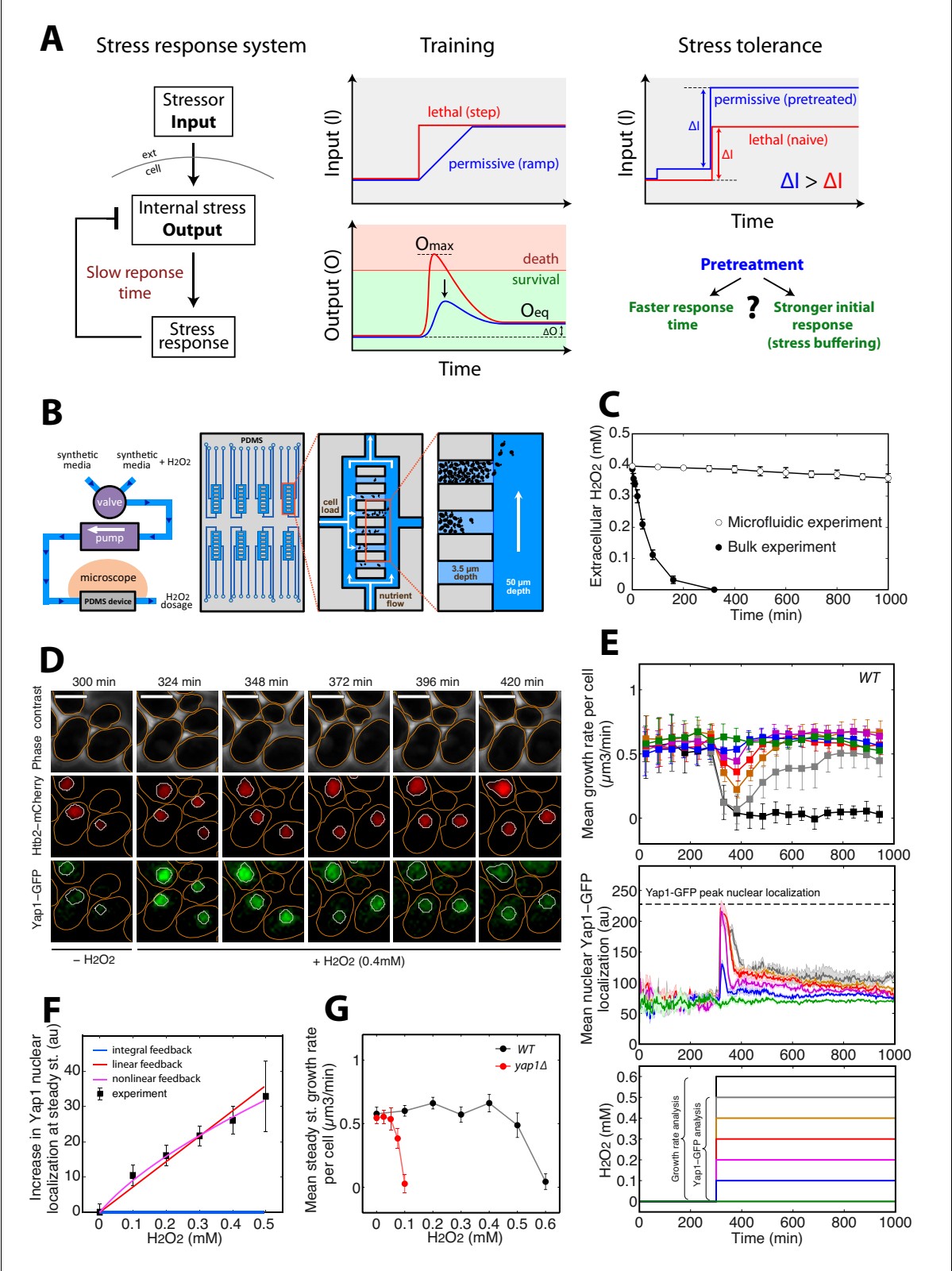

**Figure 1.** A single-cell microfluidics assay to monitor yeast adaptation to $H_2O_2$. (**A**) Schematic representation of 'training' and 'stress tolerance' phenomena in a simple negative feedback-based system. (**B**) Schematic of the microfluidic device setup for live-cell imaging in $H_2O_2$-containing media. (**C**) Decline in extracellular $H_2O_2$ concentration comparing the single-cell assay and bulk experiment (starting at cellular OD600 = 0.5). (**D**) Sequence of phase-contrast and fluorescence images of cells at the indicated time points after addition of 0.4 mM $H_2O_2$ at t = 300 min. The red and green channels

*Figure 1 continued on next page*

*Figure 1 continued*

represent the Htb2-mCherry (nuclear marker) and Yap1-GFP signals, respectively. Orange and white lines represent the cellular and nuclear contours obtained after automated segmentation. The white bars represent 5 μm. (E) Top: Mean growth rate per cell as a function of time after addition of different $H_2O_2$ concentrations at t = 300 min, as indicated in the bottom panel. Middle: Mean nuclear Yap1-GFP localization as a function of time, with the same color coding as in the top and bottom panels. Yap1-GFP scoring is not possible at 0.6 mM due to metabolic arrest/cell death and GFP signal decline. (F) Increase in mean Yap1-GFP localization (relative to the pre-stress level) at steady state (measured for t > 800 min) as a function of $H_2O_2$ concentration added at t = 300 min during a step experiment. Lines represent the best fits to mathematical models of integral (blue), linear (red) and nonlinear (magenta) feedback with different sets of assumptions (see Materials and methods). (G) Comparison of growth rate at steady state as a function of $H_2O_2$ concentration for the wild-type (WT) strain and the *Δyap1* mutant. Error bars and shaded regions are SEM (C, N = 6; E, N > 100 for most time points; F and G, N > 100 for each $H_2O_2$ concentration). See also *Figure 1—figure supplement 1* and Materials and methods.

The following figure supplements are available for figure 1:

**Figure supplement 1.** Medium diffusion properties in the microfluidics device.

**Figure supplement 2.** Principle of growth rate measurements.

to decipher the mechanisms at work in *adaptive homeostasis*. We developed a quantitative framework based on mathematical modeling, live-cell imaging, and microfluidics to apply controlled $H_2O_2$ stress patterns and to measure the adaptive stress response with a high level of accuracy. We demonstrate that cell survival is highly stress-rate dependent, validating the paradigm that adaptation is limited by the response time of the cellular homeostatic machinery and thus revealing an unprecedented 'trainability' of the cells to stress. We also show that the acquisition of stress resistance is distinct from cellular training and originates in the nonlinear scavenging of $H_2O_2$ by peroxiredoxins (Prx). Along with this, our study unravels an important Prx-dependent replicative lifespan extension in the presence of very low doses of $H_2O_2$. This reveals an unprecedented direct hormetic effect that can be quantitatively accounted for by the nonlinear Prx feedback model. Our study provides the first quantitative and mechanistic analysis that establishes the link between the architecture of a fundamental stress homeostatic system and the emergence of distinct nonintuitive properties, namely cellular training to stress and *adaptive homeostasis*.

## Results

### A quantitative assay to measure adaptation to $H_2O_2$ stress in single dividing yeast cells

We first sought to characterize the kinetics of the cellular response to stepwise exposure to increasing $H_2O_2$ concentrations and the emergence of cellular adaptation. Adaptation to $H_2O_2$ is usually measured by quantifying the fraction of surviving cells after the addition of a bolus of $H_2O_2$ to cells in culture. Under these conditions, cells rapidly degrade the $H_2O_2$ in the medium and, therefore, adaptation is partly due to the removal of the stressor. To circumvent this problem and to allow precise control of the stressor level, we developed custom microfluidic device to monitor the divisions of individual cells in trapping cavities while controlling external $H_2O_2$ concentrations by continuous replenishment of $H_2O_2$-containing medium (*Figure 1B and C* and Materials and methods). Using a fluorescent dye, we checked that diffusion in the trapping cavities was not impaired by the presence of a large cell cluster in the cavity (*Figure 1—figure supplement 1A and B*) and we also verified that cell growth was not affected despite the increasing cell confinement during a typical experiment (up to 1000 min, see *Figure 1—figure supplement 2A*). Using this technique, therefore, we could follow cellular adaptation by monitoring activation of the $H_2O_2$ stress response and real-time cellular growth rate.

The signaling network activated in response to $H_2O_2$ is centered on the Yap1 regulon (*Kuge and Jones, 1994*; *Lee et al., 1999*). Yap1 is a transcription factor that, in the absence of $H_2O_2$, shuttles between the cytoplasm and nucleus. Upon exposure to $H_2O_2$, Yap1 is oxidized and nuclear export is prevented (*Azevedo et al., 2003*; *Kuge et al., 1997*; *Lee et al., 1999*). Hence, nuclear accumulation of Yap1 is a sensitive reporter of internal $H_2O_2$ levels (*Toledano et al., 2004*). In the nucleus, Yap1 activates the expression of genes involved in redox homeostasis and $H_2O_2$ scavenging (*Gasch et al.,*

*2000*; *Godon et al., 1998*; *Lee et al., 1999*), leading to a negative feedback regulatory loop (*Toledano et al., 2004*). We monitored the dynamics of individual cells expressing Yap1-GFP and a nuclear marker (histone Htb2-mCherry) with a 3-min interval. Following the switch to medium containing 0–0.4 mM $H_2O_2$ (at t = 300 min; *Figure 1D* and *Video 1*), the cells experienced a transient stress level-dependent reduction in growth rate (see Materials and methods for the principle of the measurement) followed by a complete recovery, revealing intrinsic cellular adaptation (*Figure 1E*). In parallel, the switch to $H_2O_2$ induced an abrupt burst in Yap1-GFP nuclear localization, which saturated above 0.2 mM $H_2O_2$. Similar experiments performed with higher temporal resolution (30 s interval) revealed that the nuclear relocation time is around 120 s (see *Figure 1—figure supplement 1C*), which is much faster that the overall adaptation timescale (~45–100 min, see *Figure 1E*: Yap1-GFP quantification). However, this stands significantly higher than the timescale of stressor diffusion across cavity (see *Figure 1—figure supplement 1A and B*), therefore revealing that the activation of Yap1 is set by the diffusion of the stressor across the cell membrane rather than by limited diffusion in the device.

The burst of nuclear Yap1-GFP relocation was followed by partial recovery to a steady-state level that was also $H_2O_2$ concentration-dependent (*Figure 1D–F* and *Video 2*), suggesting that Yap1 activity is still required in the adapted state (i.e. following growth recovery). Consistent with this, deletion of Yap1 strongly decreased the cells' capacity to adapt, in agreement with previous findings (*Inoue et al., 1999*) (*Figure 1G* and *Figure 1—figure supplement 2E*). Last, the fact that a 0.1 mM $H_2O_2$ stress induces a partial nuclear relocation of Yap1-GFP but has no effect on growth rate indicates that Yap1 signaling is more sensitive to $H_2O_2$ than is the overall cellular physiology (*Figure 1E*).

## $H_2O_2$ effects on physiology are mediated by sequential thresholds

Interestingly, while the cells adapted to 0.3 mM $H_2O_2$ (complete mean growth rate recovery), exposure to 0.6 mM $H_2O_2$ induced full growth arrest (*Figure 1E* and *Video 3*), revealing that cell fate is controlled by a sharp threshold in external $H_2O_2$ level. To determine the physiological changes accompanying this switch from adaptation to arrest, we focused on the behavior of individual cells exposed to a sublethal dose (0.5 mM) of $H_2O_2$ (see *Figure 2A and B*). Notably, while all cells experienced at least a transient growth arrest (at t = 390 min, *Figure 2B*) following exposure to the stressor (at t = 300 min), we observed a high degree of heterogeneity in cell fate across the population at steady-state: 22% of cells present at the time of $H_2O_2$ addition recovered a normal growth and division rate at t > 600 min (referred to as the 'adapted' phenotype in the following); 36% experienced a prolonged slow-down of cell cycle progression characterized by an extended budded period, but continued to increase in size over time ('prolonged cell cycle arrest' phenotype); and 42% stopped growing and failed to divide ('permanent growth arrest' phenotype) (*Figure 2A–C* and *Figure 2—figure supplement 1* and *Video 4*). We checked this heterogeneity in cell fate was not dependent on the position of cells with respect to the border of the cavity (see *Figure 1—figure*

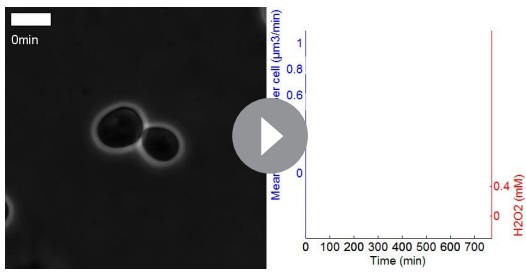

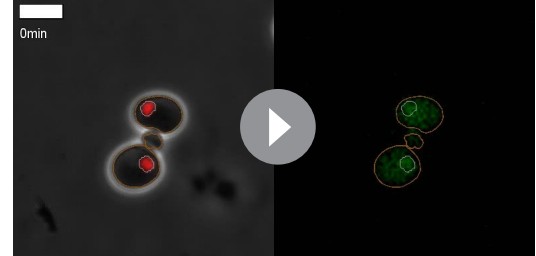

**Video 1.** Growth rate monitoring upon $H_2O_2$ stress (refers to *Figure 1*) Movie showing a time-lapse experiment where cells are exposed to sudden step stress of 0.4 mM $H_2O_2$ at t = 300 min. Left: phase contrast, right: growth rate evolution graph. The white bar represents 5 μm.

**Video 2.** Yap1 nuclear relocation upon $H_2O_2$ stress (refers to *Figure 1*) Movie showing the nuclear enrichment of Yap1 in cells exposed to 0.4 mM $H_2O_2$ at t = 300 min. Left: Phase contrast and mCherry (Htb2-mCherry) channels. Right: GFP (Yap1-GFP) channel. The white bar represents 5 μm.

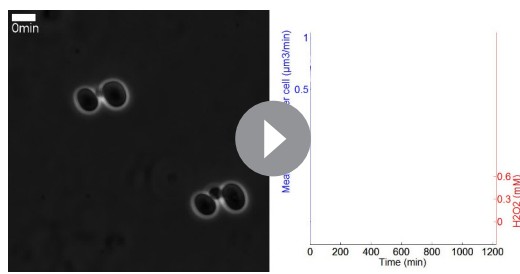

**Video 3.** Cell metabolic arrest at high $H_2O_2$ dose (refers to *Figure 1*) Movie showing a permanent cell growth arrest in cells exposed to 0.6 mM $H_2O_2$. Left: phase contrast, right: growth rate evolution graph. The white bar represents 5 μm.

supplement 1D). Interestingly, performing the same analysis of phenotypic distribution at various $H_2O_2$ levels revealed that the cell cycle arrest phenotype emerged at lower concentrations of $H_2O_2$ than the growth arrest phenotype (typically ~0.2 and 0.5 mM $H_2O_2$, respectively; *Figure 2C*).

To go beyond the characterization of growth and division impairments in cells submitted to $H_2O_2$, we asked whether these defects were accompanied by the activation of DNA damage response and/or ultimately lead to cell death. To this end, we examined the response of cells expressing Rnr3-GFP or Ddc2-GFP fusion proteins. Rnr3 is a subunit of ribonucleotide reductase and is upregulated during the DNA damage response (DDR), and Ddc2 is a DNA damage checkpoint protein that forms foci upon recruitment to DNA lesions. Exposure to 0.4 mM $H_2O_2$ substantially increased cytoplasmic Rnr3-GFP levels in cells displaying the prolonged cell cycle arrest phenotype, but not in adapted cells (*Figure 2D*). Similarly, the vast majority of cell-cycle-arrested cells displayed bright foci of Ddc2-GFP fluorescence, unlike adapted cells (*Figure 2E*). These experiments indicated that $H_2O_2$-induced DNA damage and subsequent DDR checkpoint activation were responsible for prolonged cell cycle arrest. In contrast, cells experiencing a permanent arrest showed no detectable increase in Rnr3-GFP (not shown). We considered that these cells may have a compromised physiological state preventing any response to the oxidative stress threat, and may ultimately die. To test this hypothesis, first, we used a vital stain (propidium iodide, PI) to monitor the onset of death in cells abruptly exposed to 0.6 mM $H_2O_2$ (see *Figure 2—figure supplement 2A and B* and *Video 5*). We found that all cells (N = 123) eventually became fluorescent, therefore demonstrating that a stress exposure to this $H_2O_2$ concentration ultimately induces cell lysis. Next, we wondered to which extent this growth arrest phenotype could be reverted by stress removal. To this end, we monitored the mean growth rate and the expression of the thioredoxin promoter fused to both sfGFP and a destabilizing degron sequence, TRX2pr-sfGFP-deg (*TRX2* encodes a Yap1-regulated thioredoxin), after exposure of cells to 0.6 mM $H_2O_2$ for varying periods. $H_2O_2$ addition induced rapid growth arrest and irreversible decay of TRX2pr-sfGFP-deg levels (black lines on *Figure 2—figure supplement 2C* upper and middle panels, respectively). However, removal of the stress by switching back to $H_2O_2$-free medium at various times after $H_2O_2$ addition led to recovery of the mean cellular growth rate and induced reactivation of the Yap1 regulon if the duration of exposure was less than 4 hr (*Figure 2—figure supplement 2C*). This indicated that a few hours of exposure at 0.6 mM $H_2O_2$ were necessary to induce an irreversible growth arrest phenotype that ultimately lead to cell death.

Last, we checked that the behavior of permanently arrested cells, even at sublethal $H_2O_2$ concentrations (0.4 mM $H_2O_2$), was similar to the irreversible growth arrest phenotype at 0.6 mM $H_2O_2$. For this, we compared the expression of the TRX2-GFP fusion protein following either a 0.4 mM or a 0.6 mM $H_2O_2$ step. In both cases, we observed that the subpopulation of permanently arrested cells showed very low TRX2-GFP levels (*Figure 2—figure supplement 3A–C*). In addition, we found that, unlike adapted and cell cycle arrested cells, cells with a permanent growth arrest displayed the same bright and large vacuole phenotype (*Figure 2—figure supplement 3A–C*) in both conditions (0.4 mM and 0.6 mM $H_2O_2$). Therefore, these results indicate that the permanent growth arrest phenotype observed in step experiments (lethal or sublethal) reflects the inability of the cells to defend against the stressor.

Collectively, our analysis reveals the existence of distinct cell fates following the exposure to acute sublethal doses of $H_2O_2$, and demonstrates that these phenotypes occur in a $H_2O_2$ concentration-dependent manner.

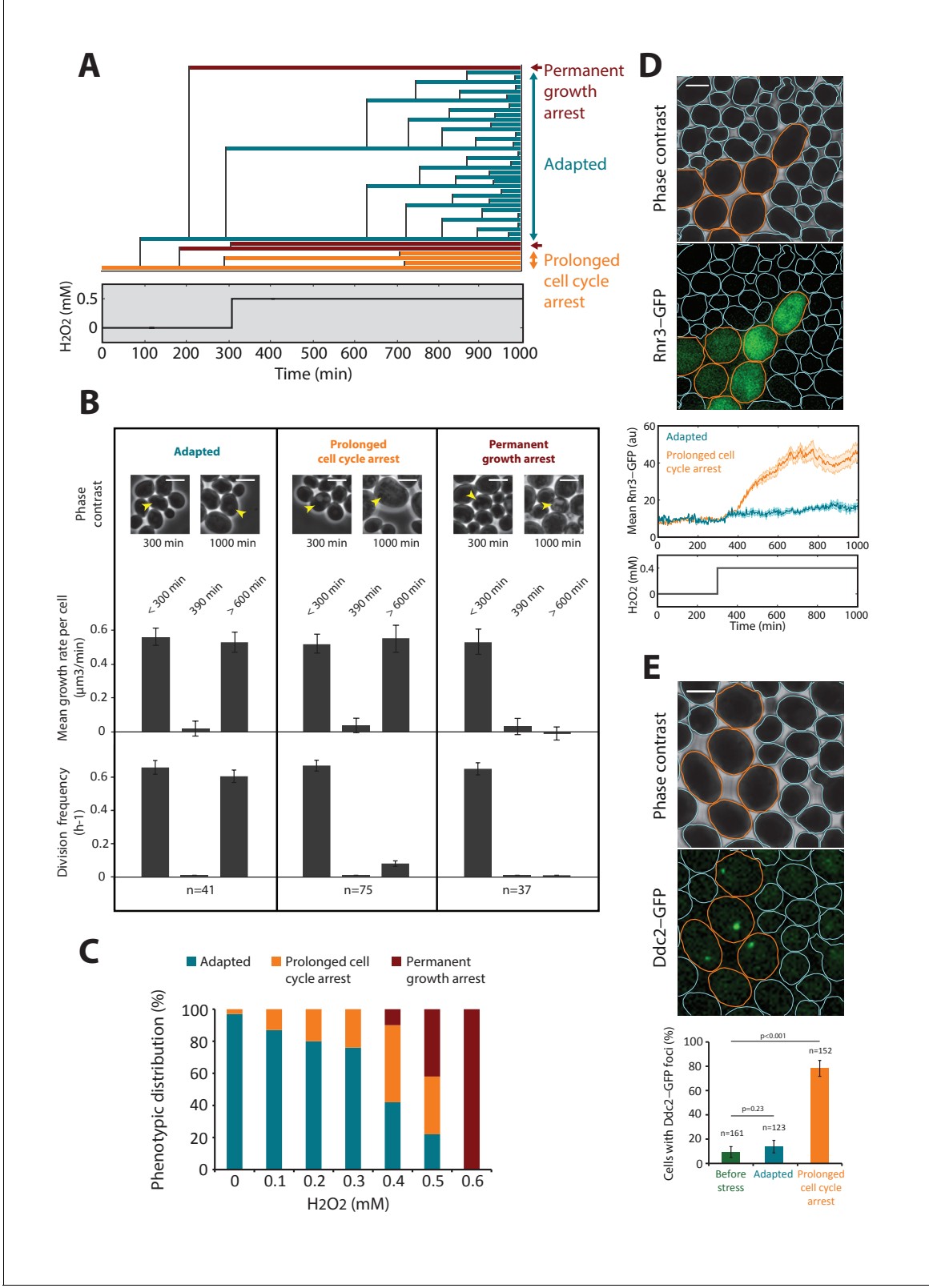

**Figure 2.** DNA damage checkpoint and metabolic arrest during exposure to $H_2O_2$. (A) Lineage of cells after addition of 0.5 mM $H_2O_2$ at t = 300 min. Each colored line corresponds to a single cell, and cell budding is indicated by the vertical black lines. The plot shows examples of cells with three distinct growth phenotypes, as indicated. (B) Quantification of the growth rate and division rate of the cellular phenotypes displayed in (A). Error bars are SEM. (C) Relative distribution (%) of cellular phenotypes as a function of $H_2O_2$ concentration added at t = 300 min. N = 100 for each concentration.
*Figure 2 continued on next page*

*Figure 2 continued*

(**D**) Top: Upregulation of cytoplasmic Rnr3-GFP expression at 300 min after addition of 0.4 mM $H_2O_2$. Bottom: Quantification of the Rnr3-GFP signal after addition of 0.4 mM $H_2O_2$. (**E**) Top: Upregulation of Ddc2-GFP expression at 300 min after addition of 0.4 mM $H_2O_2$. Bottom: Percentage of cells with durable Ddc2-GFP foci measured before and at 300 min after addition of 0.4 mM $H_2O_2$ for the cell phenotypes identified in (**A**). Error bars are 95% confidence intervals (CI). Two-proportions Z test. The whites bar represent 5 μm. See also *Figure 2—figure supplement 1*.

The following figure supplements are available for figure 2:

**Figure supplement 1.** Lineage of cells following the exposure to 0.5 mM $H_2O_2$.

**Figure supplement 2.** Cell death quantification during high (0.6 mM $H_2O_2$) step stress.

**Figure supplement 3.** Cell death markers for mild (0.4 mM $H_2O_2$) step stress.

## The kinetics of temporal stress patterns controls cellular adaptation to $H_2O_2$ through a training mechanism

The observation that increasing $H_2O_2$ external levels resulted in a sharp and partially reversible transition from adaptation to growth arrest suggested that the transient internal stress level $H_{max}$ reached closely after exposure to $H_2O_2$ may exceed a toxic concentration $H_{tox}$ beyond which cellular function is impaired, as hypothesized in the introduction. If so, exposure of cells to a gradual increase in $H_2O_2$ concentration should allow the cells more time to activate the antioxidant response and should thus dramatically improve cellular adaptation (see *Figure 1A*). To understand quantitatively how the kinetics of a stress pattern may influence cellular adaptation, we first developed a mathematical description of the homeostatic machinery based on the negative feedback regulation in the Yap1 network. This model was then used throughout this study to help identify and formalize the emergent properties of this system through iterative cycles of predictions and experimental challenges, rather than to perform exhaustive data fitting aimed at retrieving individual parameter values.

The model assumed that nuclear relocation of Yap1 increases the production of antioxidants (referred to as 'A' in the model, see *Figure 3A*), which then scavenge intracellular $H_2O_2$ ('H' in *Figure 3A*). For the sake of simplicity, we first developed a linear version of this model (f(H)=1 and g(H) = H; *Figure 3A*) that could successfully recapitulate the limited accuracy of the homeostatic system; the internal $H_2O_2$ level at steady-state $H_{eq}$ increases with the magnitude of $H_2O_2$ steps, unlike a system based on an 'integral' feedback regulatory scheme (*Figures 1E, F* and *3B*, and Materials and methods). This property is a direct consequence of the assumption that antioxidants are not infinitely stable but must be diluted in growing cells (μ'≠0, see Materials and methods and *Figure 3—figure supplement 1A and B*). However, the growth rate does not affect the kinetics of the internal $H_2O_2$ burst during the transient response to $H_2O_2$ steps (see *Figure 3—figure supplement 1* and Materials and methods). Therefore, the observation that the growth rate undergoes a transient slowdown during the regime that precedes adaptation to sublethal $H_2O_2$ steps should not impact the overall internal $H_2O_2$ kinetics nor the cellular

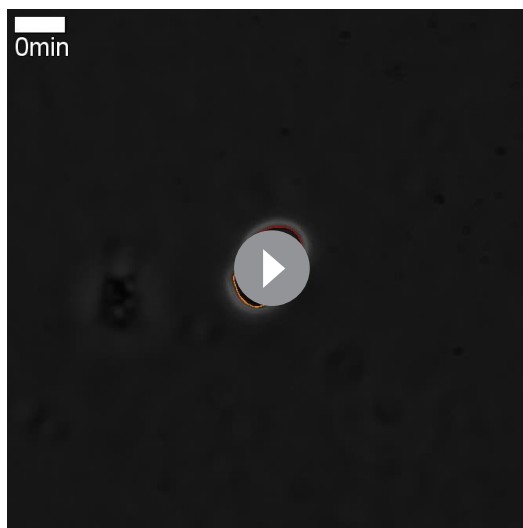

**Video 4.** Phenotypic variability upon $H_2O_2$ stress (refers to *Figure 2*) Movie showing different cell fates in cells exposed to 0.5 mM $H_2O_2$ at t = 300 min. Blue cell contours: Adapted cells; Yellow cell contours: Prolonged cell cycle arrest; Red contours: Permanent growth arrest. The white bar represents 5 μm.

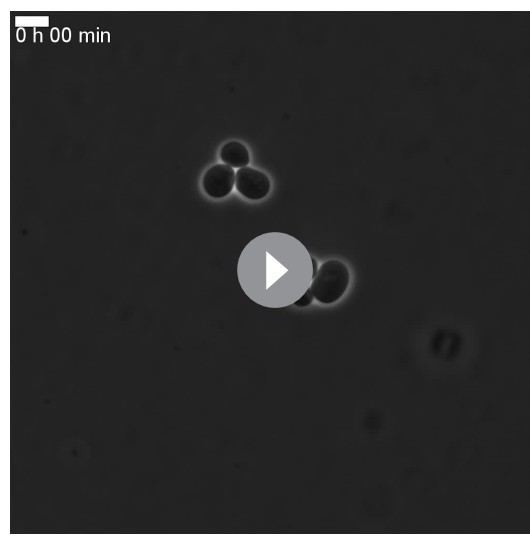

0 h 00 min

**Video 5.** Cell mortality at high $H_2O_2$ step stress (refers to **Figure 2—figure supplement 2**) Movie showing the incorporation of PI in cells exposed to 0.6 mM $H_2O_2$ at t = 300 min. The white bar represents 5 μm.

adaptation capacity. Therefore, for the sake of simplicity, we neglected the variations in growth rate in the model.

Importantly, the linear model also predicted that, whereas steep ramps would quickly lead to growth arrest (since, eventually $H_{max} > H_{tox}$; green line on **Figure 3C** and inset), as observed experimentally beyond 0.6 mM, slower stress ramps may allow the cells to adapt to much higher stress levels (keeping $H_{max} < H_{tox}$; blue lines on **Figure 3C** and inset). The phase diagram in the $(\delta, I)$ space recapitulated these predictions and, additionally, clearly delimited the region in which adaptation to high-stress magnitude $I$ is permitted (indicated by the black line $H_{max} = H_{tox}$ on **Figure 3C**). The extent of this region is limited by the overall accuracy of the homeostatic system, which is mostly set by the degradation rate $\mu'$ of antioxidant enzymes (**Figure 3—figure supplement 1E and F** and Materials and methods) and the response time of the antioxidant system, which derives from the degradation rate $\mu$ of corresponding mRNAs (**Figure 3—figure supplement 1G–J** and Materials and methods); $\mu$ was set to $\log(2)/40$ $\text{min}^{-1}$, according to previous measurements (**Geisberg et al., 2014**).

To test the prediction of the linear model, we developed a protocol to generate a linear increase in $H_2O_2$ concentration over time in the microfluidic chip, with rates from 1.1 μM/min to 17.6 μM/min (**Figure 3—figure supplement 2A and B** and Materials and methods). Under these conditions, we observed no decline in growth rate up to >4 mM $H_2O_2$ when the slope $\delta$ was $\leq 4.4$ μM/min (**Figure 3D and E**), whereas the growth rate decreased progressively at $\delta \geq 8.8$ μM/min (**Figure 3E**). We checked that this decay in growth rate observed at the population level mainly resulted from a progressively increasing number of individual cells undergoing a permanent growth arrest, consistently with the phenotype described in step experiments (**Figure 3—figure supplement 2C**). Similarly, these arresting cells displayed a high-phase-contrast intensity due to large vacuoles, suggesting that these cells are unable to adapt and ultimately die (**Figure 3—figure supplement 2E**). In addition, we verified that the growth rate decay was not due to the higher absolute $H_2O_2$ concentration reached during a 8.8 μM/min ramping experiment compared to a ramp of 4.4 μM/min. Indeed, we observed that the onset of occurrence of the growth arrest phenotype at 8.8 μM/min occurred at a lower $H_2O_2$ level that the one reached at 4.4 μM/min, whereby no growth arrest was observed (**Figure 3—figure supplement 2D**). Altogether, these results confirmed the prediction of the model that adaptation is strongly stress-rate dependent and validated the hypothesis that the transient internal $H_2O_2$ peak level reached with stepwise addition limits the ability to adapt. In agreement with this, nuclear localization of Yap1 was lower during ramping (**Figure 3E**) than during the step experiments (**Figure 1E**).

Interestingly, the model also predicted the existence of an absolute $H_2O_2$ level $I_{abs}$ beyond which no adaptation is possible, even with extremely slow ramping, due to the dilution of antioxidants (**Figure 3C**, **Figure 3—figure supplement 1**, and Materials and methods). To estimate this threshold experimentally, we monitored the growth rate of cells upon exposure to a slow $H_2O_2$ ramp ($\delta = 2.2$ μM/min) for >5000 min and found that the onset of growth decline occurred at a $H_2O_2$ concentration of 7.2 mM (**Figure 3—figure supplement 2F**). From a theoretical viewpoint, this concentration is a fundamental constant that characterizes the overall buffering capability of the homeostatic system and integrates most of the parameters of the model (see Materials and methods). Based on this estimate, combining all step and ramp experiments in the $(\delta, I)$ phase space (using growth rate as a readout of adaptation) provided good agreement between the experiments and the model (**Figure 3C and F**).

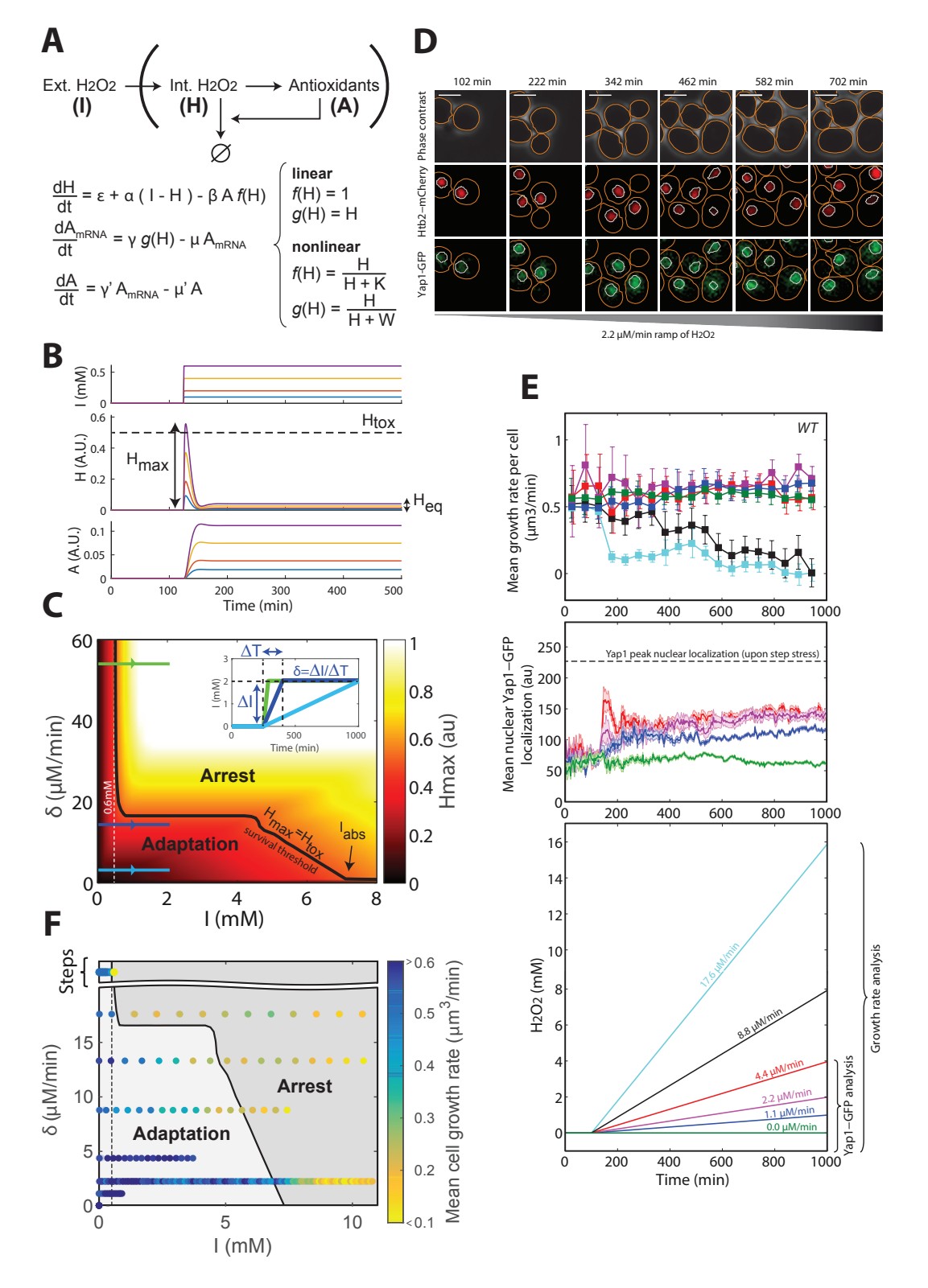

**Figure 3.** A negative feedback-based model to describe adaptation to $H_2O_2$. (**A**) Schematic of the regulatory network involved in $H_2O_2$ scavenging: external $H_2O_2$ (represented by I), internal $H_2O_2$ (represented by H), and antioxidants (represented by A). A linear set of differential equations is used to describe the evolution of this system over time. (**B**) Response of the linear system described in panel (**A**) to sudden exposure to external $H_2O_2$ (step of amplitude I). Each colored line corresponds to a given concentration of $H_2O_2$. $H_{eq}$ is the steady-state internal H concentration, $H_{max}$ is the maximum H

*Figure 3 continued on next page*

*Figure 3 continued*

concentration reached during the transient regime. $H_{tox}$ is the threshold concentration beyond which growth/division is assumed to stop (obtained for I = 0.6 mM). (**C**) Phase diagram showing $H_{max}$ as a function of the amplitude of the step I and the rate of the $H_2O_2$ ramp $\delta = \Delta I/\Delta T$. Inset shows a graphical representation of these parameters. The solid black line indicates the contour given when $H_{max} = H_{tox}$, assuming the general assumptions of the linear model described in panel A. (**D**) Sequence of phase-contrast and fluorescence images of individual cells at the indicated times after initiation (t = 100 min) of a linear ramped increase in $H_2O_2$ concentration at a rate $\delta$ of 2.2 µM/min. The red and green channels represent the Htb2-mCherry and Yap1-GFP signals, respectively. The white bars represent 5 µm. (**E**) Top: Mean growth rate per cell as a function of time after initiation (t = 100 min) of linear ramps in $H_2O_2$ concentration. The line colors correspond to the indicated ramp slopes in the bottom panel. Middle: Mean nuclear Yap1-GFP localization. Error bars and shaded regions are SEM, N > 100 for most time points. (**F**) Phase diagram recapitulating the mean growth rate of cells during adaptation to steps (*Figure 1E*) and linear ramps at various rates $\delta$. The gray shading delimits the regions of adaptation and arrest, as expected from the linear feedback model. See also *Figure 3—figure supplement 1* and Materials and methods.

The following figure supplements are available for figure 3:

**Figure supplement 1.** Linear model: response to step, ramps, and training capabilities.

**Figure supplement 2.** Experimental setup to generate $H_2O_2$ ramps and scoring of the permanent growth arrest phenotype.

Overall, this analysis identified a > 10-fold increase in the $H_2O_2$ adaptation limit observed when using slow versus fast stress ramping, which could be explained by a linear negative feedback model in which the response time of antioxidant expression plays a critical role. The unprecedented analysis therefore revealed the 'training' capabilities of individual yeast cells, which can be progressively acclimated to increasingly high levels of stress.

## Acquisition of tolerance to $H_2O_2$ is distinct from cellular training

Next, we considered how our framework based on a linear feedback model could explain the phenomenon of acquired stress tolerance, in which mild ($I_0$ = 0.1–0.4 mM) $H_2O_2$ pretreatment increased by several orders of magnitude the fraction of cells surviving a subsequent challenge with a more severe stepwise stress of magnitude $\Delta I$ (*Davies et al., 1995*). To transpose these observations using our methodology, we first verified that pretreating cells with $I_0$ = 0.2 mM $H_2O_2$ shifted the adaptation threshold to $\Delta I$ = 1 mM (*Figure 4A and B*), contrasting with the $\Delta I$ = 0.6 mM threshold obtained for $I_0$ = 0 mM (as shown in *Figure 1*). Here again, the large vacuole phenotype obtained with $\Delta I$ = 1 mM (as measured using phase-contrast intensity of cells) strongly suggested that these permanently arrested cells failed to adapt, as in steps and ramp experiments (*Figure 4—figure supplement 1A*). In addition, we found that this effect was clearly dependent on the Yap1 regulon, since *yap1Δ* mutants did not display acquisition of tolerance (*Figure 4—figure supplement 1B and C*). However, this increased resistance to stepwise stress exposure could not be explained by the linear feedback model, which predicted that both pretreated and naive cells should experience a similar internal peak stress during the subsequent stress challenge (*Figure 4C*). Mathematically, this results from an additivity principle, according to which the response to a perturbation $\Delta H$ (i.e. stress challenge) is independent of the response triggered by a preceding input fluctuation (i.e. preconditioning).

To explain this phenomenon of acquired stress tolerance, we first hypothesized that the pretreatment may switch the cell to an activated/adapted state capable of a much quicker transcriptional response to the subsequent challenge, as proposed previously in the context of salt cross-tolerance (*Guan et al., 2012*). To test this hypothesis, we monitored the rate of TRX2pr-GFP-deg accumulation upon exposure to a range of $H_2O_2$ steps and found a lower transcription rate of the *TRX2* promoter during the challenging step of magnitude $\Delta I$ = 0.6–0.8 mM than during the pretreatment of magnitude $I_0$ = 0.2 mM (*Figure 4D*), thus ruling out the hypothesis of quicker and/or stronger transcriptional reactivation of the homeostatic machinery.

As an alternative hypothesis, we reasoned that stress preconditioning might increase the $H_2O_2$-buffering efficiency (through higher scavenging rate) leading to a lower transient internal $H_2O_2$ level upon exposure to the subsequent stress challenge. In line with this, we found that the amplitude of the burst in Yap1 nuclear relocation decreased as a function of the pretreatment level $I_0$ (*Figure 4E and F*, and *Figure 4—figure supplement 1D*), consistent with the lower transcriptional activation of

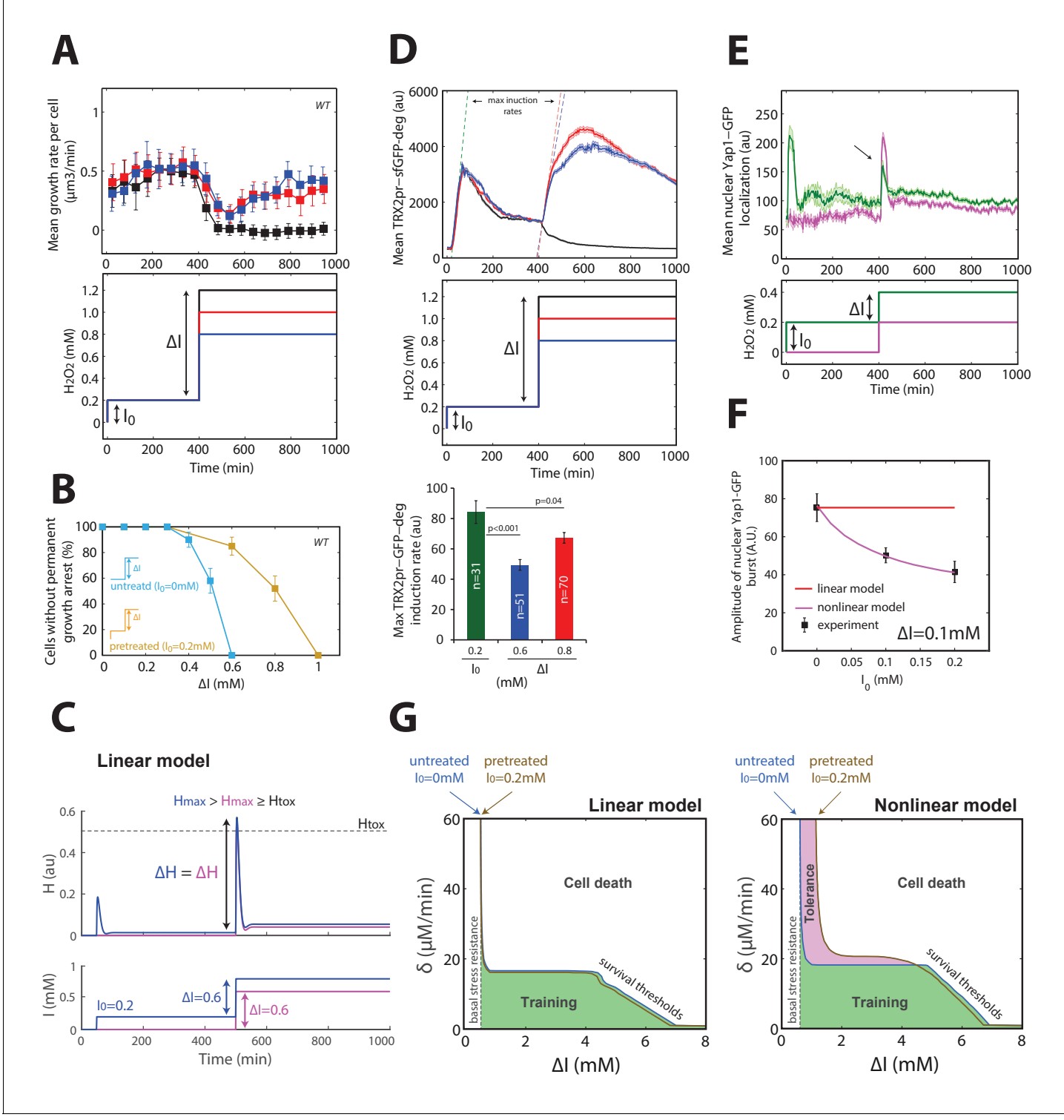

**Figure 4.** Mechanism of acquisition of tolerance to stress. (**A**) Mean cell growth rate (top panel) of cells exposed to $H_2O_2$ steps of the magnitude indicated in the bottom panel at t = 400 min after a 0.2 mM pretreatment at t = 0 min. (**B**) Fraction of cells without permanent growth arrest at different concentrations of $H_2O_2$ in the presence (yellow) or absence (blue) of pretreatment. N = 100 for each concentration. (**C**) Response of the linear system to simple $H_2O_2$ step of amplitude $\Delta I$ = 0.6 mM for naive (magenta: $I_0$ = 0 mM) or pretreated (bleu: $I_0$ = 0.2 mM) cells. (**D**) Top: Mean cell transcriptional dynamics (top panel) of the *TRX2* promoter (Trx2-sfGFP-degron) for the $H_2O_2$ treatments shown in the middle panel. Bottom: Quantification of the maximum transcription rate of the *TRX2* promoter during the indicated steps. Two-means Z test. (**E**) Mean cell Yap1-GFP nuclear localization upon a 0.2 mM $H_2O_2$ step for cells with (green line) or without (magenta line) a 0.2 mM pretreatment, as indicated in the bottom panel. (**F**) Quantification of the

*Figure 4 continued on next page*

*Figure 4 continued*

amplitude of the burst in Yap1 nuclear localization during a 0.1 mM $H_2O_2$ challenge step. The lines indicate the fit of the linear (red) and nonlinear (magenta) models. (G) Numerical phase diagram indicating the region in which adaptation is permitted as a function of the overall stress magnitude $\Delta I$ and stress rate $\delta$ for the linear (left) and nonlinear (right) models. The solid black line indicates the contour given when $H_{max} = H_{tox}$ (survival threshold) as in *Figure 3C*. The vertical dashed line represents the basal stress resistance, as observed in step experiments. The green color represents the region in which cells can be trained to resist higher stress levels through a slow ramping process. The magenta region highlights the shift in survival threshold obtained following a pretreatment according to the nonlinear model. A, D–F, error bars and shaded regions are SEM (N > 100 for most time points). B, error bars are 95% CI. See also *Figure 4—figure supplement 1*.

The following figure supplement is available for figure 4:

**Figure supplement 1.** Acquisition of tolerance: comparison of the linear and nonlinear models.

its effector gene *TRX2*. This indicated that the pretreated cells perceive a lower internal stress level than do naive cells.

To quantitatively account for these observations, we sought to refine the mathematical description of the homeostatic system. According to the linear model, the scavenging rate depends only on the concentration of antioxidant enzymes A, meaning that $H_2O_2$-scavenging enzymes would always be saturated by the $H_2O_2$ substrate following stress exposure. If, instead, we consider that the enzymes are sufficiently abundant or the internal $H_2O_2$ level is sufficiently low that enzyme saturation does not systematically occur, then the scavenging rate in the model becomes a nonlinear function of the two variables A and H (see Materials and methods). Consequently, stress pretreatment may drive the homeostatic system to an equilibrium state in which the upregulated enzymes not only function to counteract the existing $H_2O_2$ flux but may also contribute *with no delay* (i.e. before any transcriptional response) to the scavenging of a future stepwise $H_2O_2$ exposure. Thus, unlike the linear model, this nonlinear model was able to quantitatively recapitulate the clear $I_0$-dependent reduction in peak internal $H_2O_2$ during the challenge step (*Figure 4—figure supplement 1E and F*), the magnitude of which was similar to the experimentally observed Yap1-GFP nuclear relocation (*Figure 4F* and *Figure 4—figure supplement 1D*). Finally, computing the phase diagram for the nonlinear model revealed that, whereas buffering of slow external fluctuations in $H_2O_2$ levels (i.e. through cellular training) is a generic property of homeostatic systems based on negative feedback loops, the adaptation to fast fluctuations in the external stressor levels following stress preconditioning (i.e. through acquisition of stress tolerance), is a distinct property that requires a specific nonlinear scavenging model (*Figure 4G*).

## Peroxiredoxins are key components of the $H_2O_2$ homeostatic machinery

Thus far, our framework has made no assumptions regarding the nature of the scavenging enzyme(s) responsible for $H_2O_2$ degradation. Therefore, we next sought to identify which of the Yap1 regulon effectors (*Godon et al., 1998*) are critical for $H_2O_2$ homeostasis. During the step experiments, adaptation could result from parallel protective and repair mechanisms (DDR, protein quality control, metabolic control, $H_2O_2$ scavenging). However, the ramp experiments, by eliminating the cellular response triggered by high transient $H_2O_2$ levels, provided a unique framework to specifically decipher the core genes of the $H_2O_2$ homeostatic machinery.

To address this, we examined the growth rates of various mutants at 300 and 800 min after initiation of a stress ramp of $\delta$ = 1.1 µM/min (*Figure 5A*). Deletion of Yap1 abolished adaptation (*Figure 5B*) with onset of growth arrest occurring at ~0.1 mM $H_2O_2$ (*Figure 5C*), similar to the threshold observed in step experiments (*Figure 1G*). The complete absence of 'trainability' of the Yap1 mutant contrasted with the efficient adaptation of the *msn2Δ msn4Δ* mutant (*Figure 5B*), which lacks the transcription factors involved in the general stress response. Similarly, mutants lacking enzymes involved in membrane lipid biosynthesis, *erg3Δ* and *erg6Δ*, adapted perfectly (*Figure 5B*), thus ruling out the possibility that reduced membrane permeability is responsible for adaptation to $H_2O_2$ ramps (*Branco et al., 2004*).

Interestingly, we found that deletion of known $H_2O_2$ scavengers, such as the mitochondrial cytochrome c peroxidase Ccp1 or the cytosolic and peroxisomal catalases Ctt1 and Cta1, did not contribute to adaptation (*Figure 5D*). Since these mutants were previously described to be

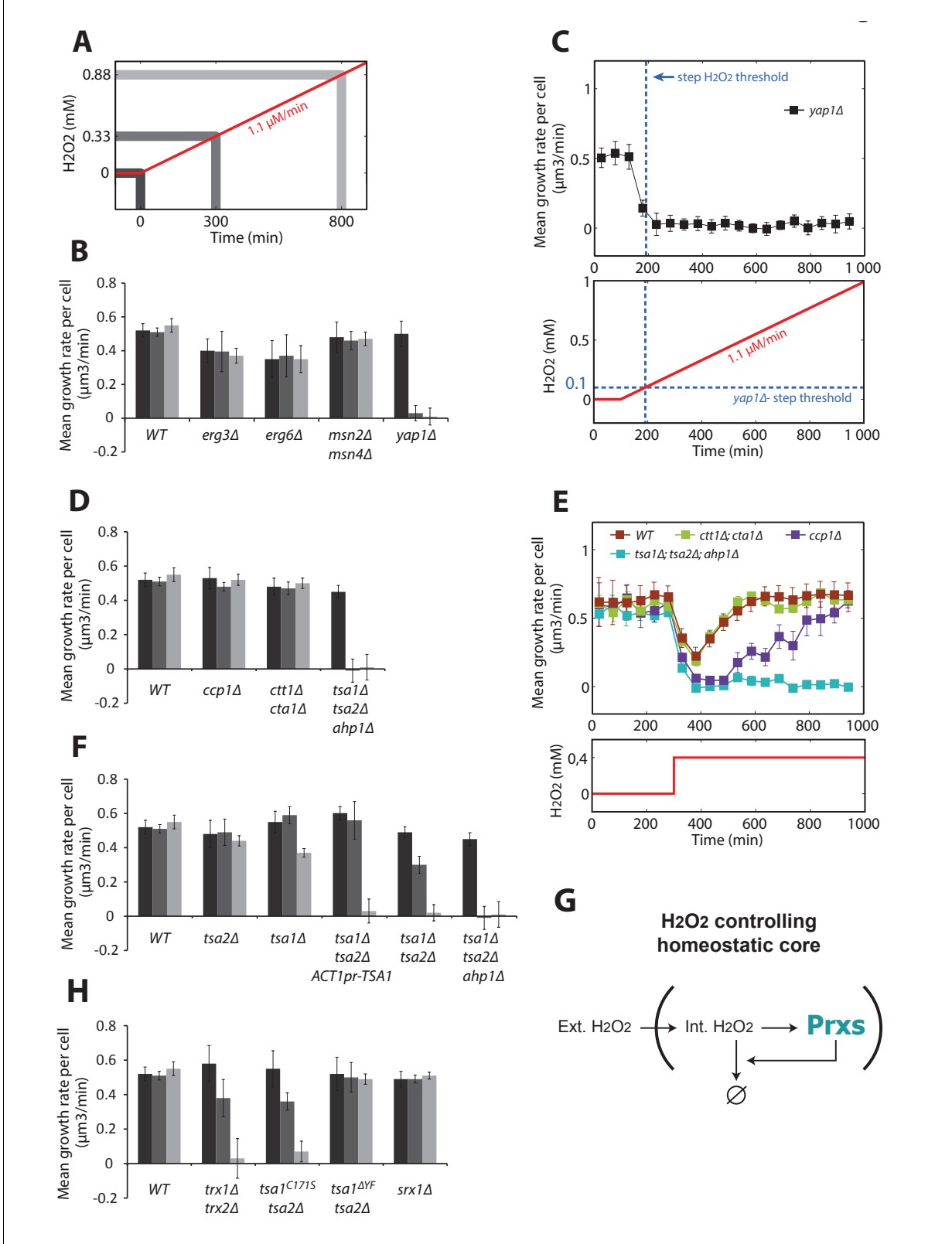

**Figure 5.** Genetic determinants of adaptation to ramped increases in H₂O₂. Quantification of mean growth rate upon exposure to a linear ramp (δ = 1.1 μM/min) starting at t = 100 min in various genotypes. (A) Illustration of the H₂O₂ ramp experiment indicating the timing of the measurements. (B) Stress response and membrane permeability mutants. (C) Details of the ramp experiment in the Δyap1 mutant. The dashed blue lines indicate the adaptation threshold obtained in step experiments (*Figure 1G*). (D) Yap1 effectors mutants. (E) Step experiment performed with Yap1 effectors mutants

*Figure 5 continued on next page*

*Figure 5 continued*

exposed to 0.4 mM H$_2$O$_2$. (**F**) Prxs mutants. (**G**) Schematic of a negative feedback control showing the essential role of Prxs in the H2O2 homeostasis. (**H**) Mutants affecting the peroxidatic cycle of Prxs. Error bars are SEM (N > 100). See also *Figure 5—figure supplement 1*.

The following figure supplement is available for figure 5:

**Figure supplement 1.** Quantification of Tsa1 expression from the *ACT1* promoter.

hypersensitive to H$_2$O$_2$ (*Jiang and English, 2006*), we wondered whether they are required for adaptation to H$_2$O$_2$ steps. Indeed, growth recovery was delayed in the *ccp1Δ* mutant (but not the *ctt1Δcta1Δ* mutant) compared with wild-type cells following exposure to a single dose of 0.4 mM H$_2$O$_2$ (*Figure 5E*). This finding indicates that some H$_2$O$_2$ scavengers may actively contribute to the transient stress response even if they are not implicated in overall H$_2$O$_2$ homeostasis. In contrast, simultaneously deleting the three peroxiredoxin genes *TSA1*, *TSA2*, and *AHP1*, which encode the two yeast 2-Cys Prxs and the atypical Prx Ahp1 (*tsa1Δtsa2Δahp1Δ*), abolished cell growth whether the cells were exposed to H$_2$O$_2$ as ramps or as steps (*Figure 5D and E*, respectively). Additionally, the extent of adaptation was proportional to the number of cytosolic Prxs present (*Figure 5F*). Constitutive expression of Tsa1 under the *ACT1* promoter (albeit less effective than the endogenous promoter, *Figure 5—figure supplement 1A and B*) in a *tsa1Δ tsa2Δ* background did not complement Tsa1 function, indicating the importance of the H$_2$O$_2$-dependent transcriptional induction of the corresponding effector genes (*Figure 5F*). Altogether, these observations demonstrate that Prxs are the essential antioxidants ensuring H2O2 homeostasis (*Figure 5G*).

2-Cys Prxs are moonlighting enzymes that reversibly switch their function from H$_2$O$_2$ scavengers to chaperones upon hyperoxidation of their peroxidatic cysteine (C$_P$) and reduction of this form by sulfiredoxin (Srx1) (*Biteau et al., 2003*; *Jang et al., 2004*). To determine which of the two Tsa1 functions is involved in adaptation to stress ramps, we first tested a mutant lacking the two cytosolic Trxs (*trx1Δtrx2Δ*) that assist Prxs in H$_2$O$_2$ scavenging, but not in protein quality control (PQC). We observed that adaptation was severely compromised in this strain (*Figure 5H*). Next, we examined a strain lacking Tsa2 and carrying a *TSA1* mutation that impairs the peroxidatic cycle of Prx but not its function in PQC (*tsa1$^{C171S}$tsa2Δ*) (*Hanzén et al., 2016*), and found that this strain also adapted poorly to H$_2$O$_2$ (*Figure 5H*). However, a strain lacking Tsa2 and carrying a *TSA1* mutation that prevents hyperoxidation and specifically impairs the enzyme's PQC function (*tsa1ΔYFtsa2Δ*) displayed a wild-type adaptation response (*Figure 5H*). Lastly, we tested the *Δsrx1* strain, in which the defective reduction of hyperoxidized Tsa1 and Tsa2 severely impairs H$_2$O$_2$ scavenging in classical techniques (*Biteau et al., 2003*) and which Prx-mediated PQC is also defective (*Hanzén et al., 2016*). Surprisingly, the *Δsrx1* strain displayed wild-type adaptation to H$_2$O$_2$ (*Figure 5H*). Taken together, this genetic analysis indicated that the peroxidatic, not the chaperone, function of 2-Cys Prxs is required for adaptation to stress ramps. The dispensability of Srx1 for adaptation suggests that the ramp protocol allows the cell to maintain internal H$_2$O$_2$ at low levels, thereby preventing Tsa1 and Tsa2 hyperoxidation.

## Tsa1 expression dynamics and nonlinear scaling with the input stress level

The mathematical model predicts that if Prx enzymes are the essential mediators of adaptation to H$_2$O$_2$, we would expect to observe strong and stable upregulation of these proteins upon exposure to H$_2$O$_2$ stress (*Figure 4—figure supplement 1F*). Indeed, a sustained increase in cytoplasmic Tsa1-GFP levels was observed upon exposure to a 0.4 mM H$_2$O$_2$ step (*Figure 6A*). This upregulation was accompanied by formation of fluorescent foci, as noted in previous studies with Tsa1-GFP (*Hanzén et al., 2016*; *Weids and Grant, 2014*). However, from the quantitative analysis of the dynamics of Tsa1-GFP protein upregulation (*Figure 6B and C*) and transcriptional activation (*Figure 6—figure supplement 1A and B*) upon H$_2$O$_2$ stress, we found that steady-state Tsa1 levels did not scale linearly, especially at low H$_2$O$_2$ concentrations (*Figure 6C* inset). Similarly, the scaling of Tsa1-GFP expression during a ramp experiment (δ = 1.1 µM/min) was sublinear (*Figure 6D*). These observations were in very good agreement with the nonlinear model, further ruling out the linear

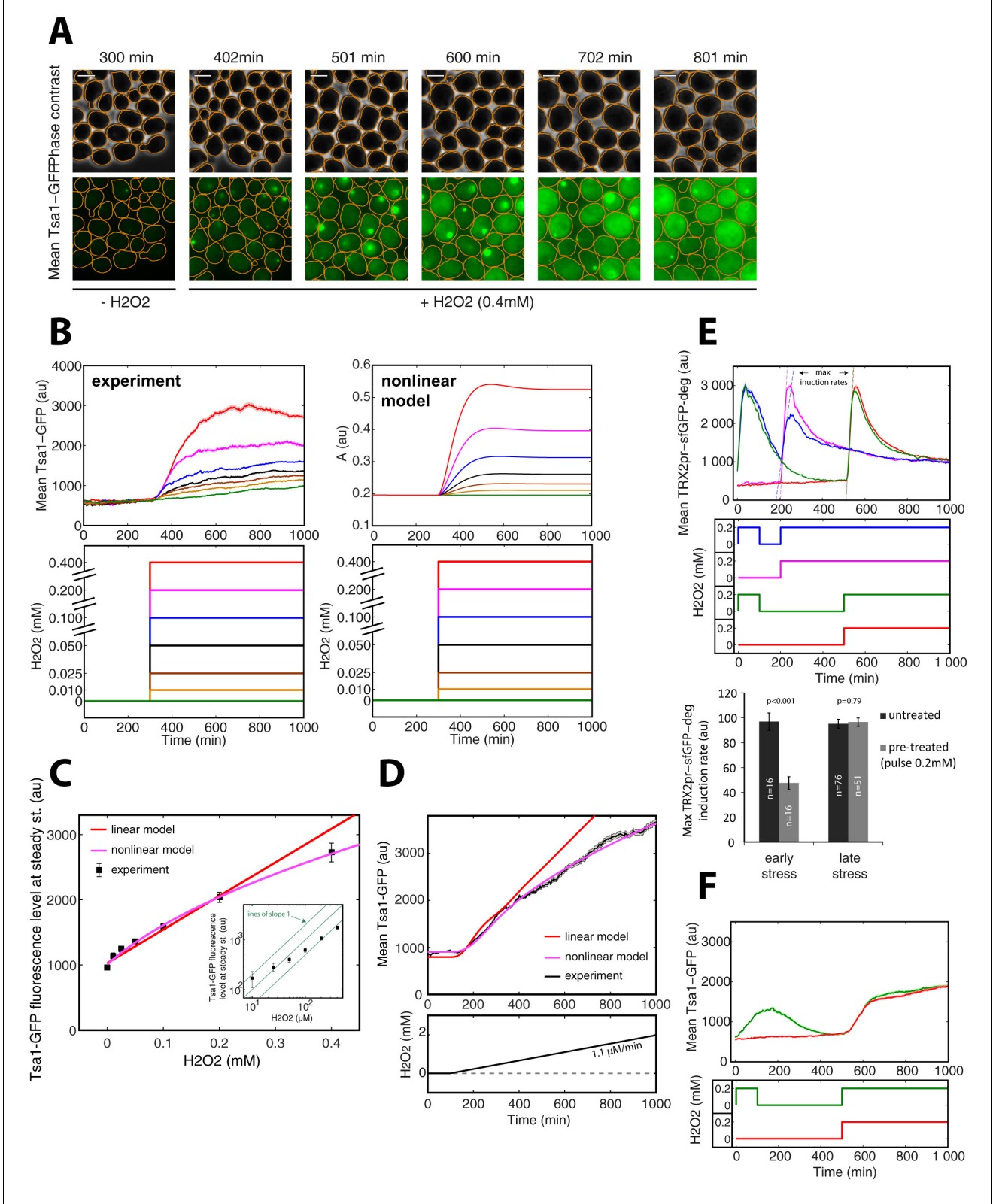

**Figure 6.** Tsa1 scaling properties. (A) Sequence of phase-contrast and fluorescence images of individual cells at the indicated times after initiation (t = 300 min) of a 0.4 mM step in $H_2O_2$ concentration. The green channel represents the Tsa1-GFP signal. The white bars represent 5 µm. (B) Left: Mean cell expression (top) of Tsa1-GFP following the $H_2O_2$ steps indicated in the bottom panel. Right: Dynamics of antioxidant level with increasing stress, as expected from the nonlinear model. (C) Quantification of mean cell expression of Tsa1-GFP at steady state as a function of $H_2O_2$ concentration (from

*Figure 6 continued on next page*

*Figure 6 continued*

experiments in (B)). Colored lines indicate the fit of the linear (red) and nonlinear (magenta) models. Inset: log-log representation of Tsa1-GFP level with $H_2O_2$ level. Green lines indicate lines of slope one on a log-log scale, to emphasize the nonlinearity of Tsa1-GFP expression. (D) Mean Tsa1-GFP expression (top) during a ramp experiment, as indicated in the bottom panel. Colored lines indicate the fit of the linear (red) and nonlinear (magenta) models. (E) Top: Mean cellular transcription of the *TRX2* promoter for cells exposed to the temporal $H_2O_2$ profiles described in the middle panel (with corresponding color coding). Bottom: Quantification of maximal transcriptional output during a step experiment performed at t = 200 min (early stress) or 500 min (late stress) with or without pretreatment. Student's t-test. (F) Mean cellular expression of Tsa1-GFP for cells exposed to the temporal $H_2O_2$ profiles described in the bottom panel. Error bars and shaded regions are SEM (B,D-F: N > 100 for most time points, C: N > 100).

The following figure supplement is available for figure 6:

**Figure supplement 1.** Analysis of the *TSA1* promoter activity during $H_2O_2$ stress.

model (*Figure 6B–D*), and the steady-state Yap1-GFP level was also best fit using the nonlinear model (*Figure 1F*). These results suggest that the $H_2O_2$ scavenging capacity of the homeostatic system becomes increasingly more efficient with the accumulation of Prxs as the stress level increases, so that Prxs do not need to be upregulated in proportion to the external $H_2O_2$ concentration.

We reasoned that, in this scenario, the stress-resistant state following pretreatment should quickly disappear as Prx enzymes are diluted upon removal of $H_2O_2$. To test this, we pretreated the cells with a 100 min pulse of 0.2 mM $H_2O_2$ and examined the transcriptional response (TRX2pr-sfGFP-deg) upon challenge 100 or 400 min later (*Figure 6E*). Notably, the transcriptional output of cells challenged 400 min after pretreatment was identical to that of cells exposed to $H_2O_2$ for the first time, indicating that the cells had returned to a naïve state by 400 min (*Figure 6E* upper panel: green vs red lines, and histogram). In contrast, TRX2pr-sfGFP transcription was lower in cells challenged 100 min after pretreatment compared with naive cells (*Figure 6E*, blue vs pink lines, and histogram). In parallel, we also observed that Tsa1-GFP levels were comparable to the basal level after a 400-min recovery period, but not after 100 min (*Figure 6F*), suggesting that Prx levels are tightly associated with the $H_2O_2$ buffering efficiency.

## A Tsa1-dependent hormetic effect of $H_2O_2$ on replicative lifespan

The nonlinear scaling of Tsa1 expression upon exposure to $H_2O_2$ may prove beneficial to cellular physiology in general, particularly during replicative aging. In support of this, the extension of both chronological and replicative longevity by caloric restriction has been shown to be mediated, at least in part, by activation of $H_2O_2$-dependent genes (*Mesquita et al., 2010*; *Molin et al., 2011*). Furthermore, recent work has shown that Tsa1 plays a role in processing of age-related protein aggregates, and overexpression of Tsa1 alone increases longevity through a mechanism involving the PQC machinery (*Hanzén et al., 2016*). To explore how activation of the $H_2O_2$ homeostatic machinery affects longevity, we measured the replicative lifespan (RLS) of cells exposed to various doses of $H_2O_2$. For this, we developed a microfluidic device that allows individual cells with different genetic backgrounds to be tracked microscopically from birth to death in separate channels (*Figure 7A and B* and *Video 6*). This is similar to our previously described device (*Fehrmann et al., 2013*), except that the large increase in capacity allows cells from up to 10 mutant strains to be tracked in parallel.

Using this technique, we observed that $H_2O_2$ had a biphasic effect on RLS. Exposure to 10 and 25 µM $H_2O_2$ significantly increased RLS up to ~25% compared with unstressed cells (median RLS generations of 26, 33, and 32 at 0, 10, and 25 µM $H_2O_2$ respectively), whereas concentrations > 50 µM $H_2O_2$ decreased RLS (median RLS of 25 and 20 generations at 50 and 100 µM $H_2O_2$, respectively; *Figure 7C and D*). This suggested that $H_2O_2$ had a hormetic effect, in which low doses of $H_2O_2$ were beneficial and improved RLS, whereas higher doses had deleterious consequences on longevity (*Ristow and Schmeisser, 2011*).

We next determined whether this effect was dependent on activation of the $H_2O_2$ homeostatic core machinery. Deletion of Yap1 or Tsa1 abolished the increase in longevity at low $H_2O_2$ doses (*Figure 7E*), implicating these proteins in the hormetic effect. We hypothesized that hormesis results from an unbalance between the substantial upregulation of Tsa1 observed at low $H_2O_2$ levels and the production of deleterious cellular damages. Indeed, the fraction of cells with foci of Ddc2-GFP (DDR marker) and Hsp104-GFP (protein aggregate marker) was similar to the unstressed control up

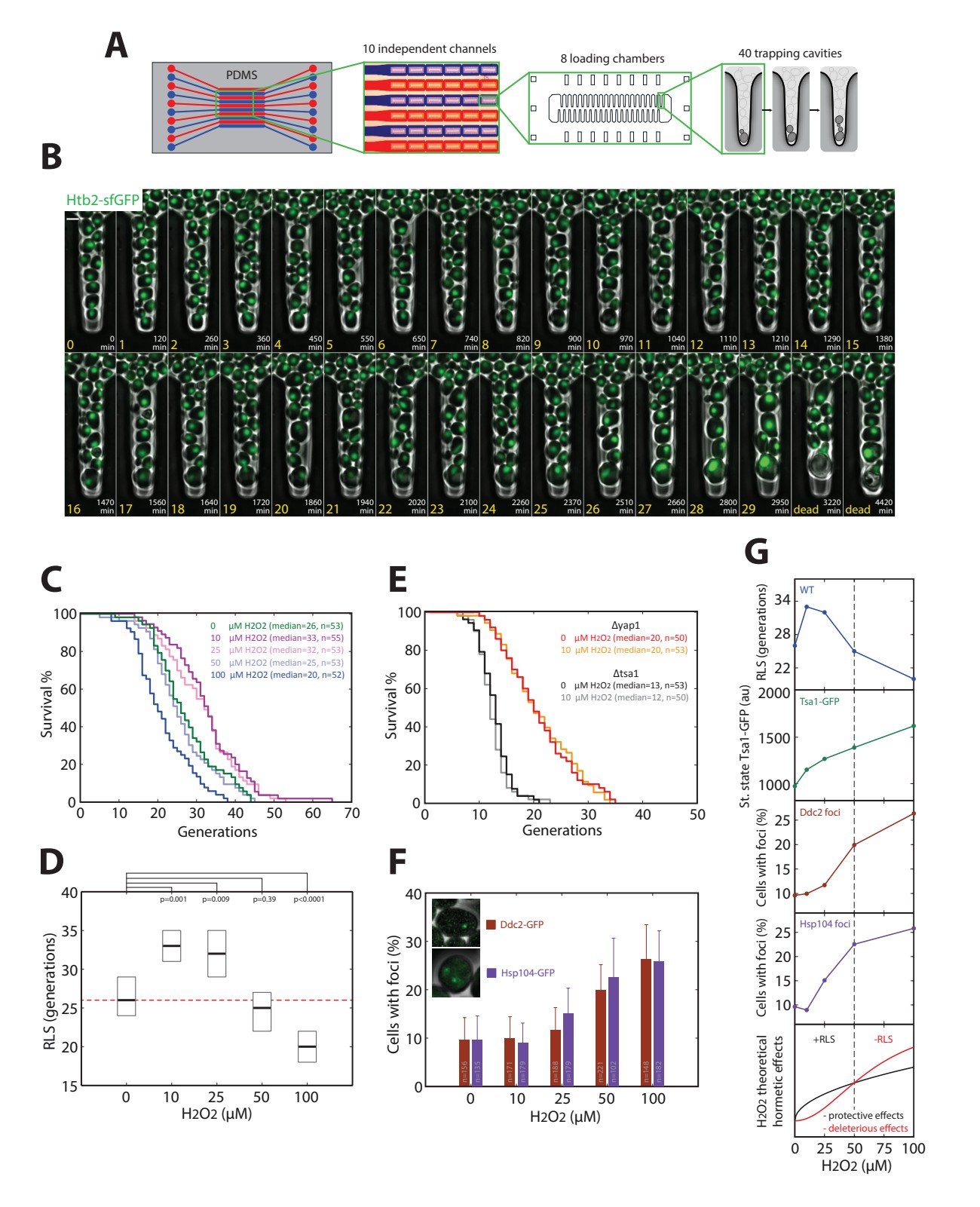

**Figure 7.** Hormetic effect of H₂O₂ on replicative longevity. (**A**) Sketch of the microfluidic device used for replicative aging experiments. (**B**) Sequence of overlaid phase-contrast and fluorescence images (Htb2-sfGFP, used as a nuclear marker to score cell division) of mother cells growing in individual cavities. Numbers indicate the timing (white) and number of cell divisions (yellow) for the mother cell at the tip of the cavity. (**C**) Survival curves for wild-type cells growing in media containing H₂O₂ at the indicated concentrations. (**D**) RLS as a function of H₂O₂ concentration. Box represents median and

*Figure 7 continued on next page*

*Figure 7 continued*

95% CI. U test. Red line shows the median RLS for 0 mM H$_2$O$_2$. (**E**) Survival curves of *Δyap1* and *Δtsa1* mutants in the presence or absence of 10 μM H$_2$O$_2$. (**F**) Frequency of specific fluorescence foci (Ddc2-GFP, Hsp104-GFP) as a function of H$_2$O$_2$ concentration. Error bars are 95% CI. (**G**) (Top to bottom) Recapitulation of measurements of RLS (blue), Tsa1-GFP steady-state upregulation (green), and frequency of damage (DDC2 foci in brown, Hsp104 foci in purple). Bottom: Conceptual sketch showing the contributions of protective (black) and deleterious (red) effects of H$_2$O$_2$ on RLS.

to 25 μM H$_2$O$_2$ (*Figure 7F and G*), while Tsa1 level experienced a ~ 25% increase in this range of concentration (*Figure 6B and C*). Above 25 μM H$_2$O$_2$, however, the increase in Tsa1 expression was relatively lower than the increase in foci (*Figure 7F and G*). Collectively, these data suggest that nonlinear activation of the homeostatic machinery in response to H$_2$O$_2$ leads to a biphasic effect on replicative longevity as recapitulated on *Figure 7G*.

## Discussion

In the present study, we combined microfluidics technology with yeast genetics and live-cell imaging to perform a comprehensive analysis of the complex adaptive properties of budding yeast to oxidative stress. The decisive element of our methodology was the precise temporal control of oxidative stress levels, without which the detailed mechanisms of adaptation to H$_2$O$_2$ stress could not have been quantitatively addressed. Indeed, it is likely that many of the discrepancies observed in studies of cellular resistance to H$_2$O$_2$ using classical bulk techniques originate from both the lack of environmental control and the population size-dependent consumption of the stressor by the cells growing in a test tube or on a plate. Our methodology provides a rationalization of H$_2$O$_2$ resistance assays, in which the final H$_2$O$_2$ concentration and its rate of increase share equal importance in determining cell fate. To date, only a few studies have used linear ramps (*Muzzey et al., 2009*; *Sorre et al., 2014*; *Young et al., 2013*) or other time-varying stimuli (*Castillo-Hair et al., 2015*) to decipher how temporal patterns of specific inputs govern activation of a regulatory network and the determination of cell fate. We anticipate that such precise dynamic environmental control will become increasingly important for refining our understanding of information processing by signaling networks.

The use of customized temporal stress patterns appears to be a unique methodology for unraveling the specific functional role of genes involved in H$_2$O$_2$ homeostasis. Our analysis has not only identified the peroxiredoxins coding genes *TSA1*, *TSA2*, and *AHP1* as key elements in ensuring adaptation to stress ramps but also revealed that other antioxidant genes, such as *CCP1*, *CTT1*, and *CTA1*, are not required under these conditions. However, *CCP1* does contribute to growth rate recovery in the step experiments. These data, therefore, suggest that Prx activity is the major determinant of H$_2$O$_2$ homeostasis and is essential regardless of the temporal stress pattern, whereas *CCP1* may only contribute to detoxification during the high transient H$_2$O$_2$ peak that accompanies acute stress. Future functional studies will be instrumental in dissecting the relative contributions of all H$_2$O$_2$-scavenging enzymes.

Beyond the necessary accuracy of a homeostatic control mechanism, our study stresses that 'trainability' is another essential functional property intimately linked to the response time of the underlying regulatory mechanism. Training has important physiological implications; namely, that cells can be acclimated to much higher stress levels (>10 fold) when the rate of increase

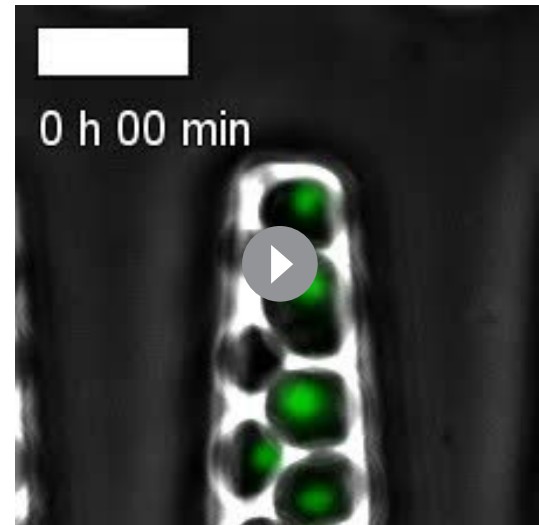

**Video 6.** Replicative lifespan monitoring (refers to *Figure 7*) Movie showing the entire lifespan of a cell from birth to death in a cavity of PDMS device. The white bar represents 5μm.

is low compared with acute exposure. The quantitative framework developed here formalizes the key parameters, such as enzyme dilution rate, that control the extent of trainability in a homeostatic system based on transcriptional regulation. In addition, the framework suggests that there is an optimal (i.e., fastest) stress pattern (following the $H_{max} = H_{tox}$ contour line on *Figure 3C*) for driving a homeostatic system to its maximal stress-buffering capability.

Interestingly, the ability to resist progressively increasing environmental insults resembles the concept of mithridatism, which was originally described as a defense strategy against poisoning, but which has lacked overall biological relevance due to the absence of similar mechanistic evidence (*Valle et al., 2012*). In the context of $H_2O_2$ stress, our study identifies a new form of mithridatism at the cellular scale that could be quantitatively explained by a mathematical description of a generic negative feedback-based regulatory network. Therefore, we envision that this framework could be transposed to other homeostatic systems and, in particular, may find potential technological applications in which improved cellular adaptation is desired without any genetic modification, such as environmental detoxification and chemotherapy.

An important outcome of our study is the establishment of a clear distinction between training and acquired stress resistance. Although both mechanisms provide a way to increase basal cellular stress resistance, their functional properties and their underlying mechanisms are completely different. Unlike training, which enables increased adaptation to slow stress fluctuations, acquired stress resistance elicits improved survival of preconditioned cells to rapid environmental changes. This latter phenomenon cannot be described using a linear model but can be explained by nonlinearity in the control loop, in which the $H_2O_2$ degradation rate depends on both the level of scavenging enzymes and internal $H_2O_2$ concentrations.

Even at basal levels, Prxs are very abundant proteins (~1% of the total dry cellular mass in mammals, [*Low et al., 2007*]), and our data show that their expression increases up to fivefold upon exposure to $H_2O_2$. Therefore, it is likely that the cellular concentration of the reduced form of these enzymes always surpasses the internal $H_2O_2$ concentration, and, consequently, the main hypothesis of our nonlinear model is that the $H_2O_2$ scavenging rate of Prxs is limited by the internal $H_2O_2$ concentration. Further biochemical studies of the entire peroxidatic cycle will be necessary to understand how and why the Prx scavenging rate depends on internal $H_2O_2$ levels *in vivo*, since other Yap1-dependent effectors, such as thioredoxins and sulfiredoxins, control the pool of reduced Prxs upon stress exposure. In the meantime, we speculate that the reason redox homeostasis relies on these extremely abundant stoichiometric enzymes rather than on catalases (which catalyze the dismutation of $H_2O_2$), as shown here, is that nonlinearity in the scavenging rate has important physiological consequences, such as stress tolerance.

The phenomenon of acquired stress resistance illustrates the plasticity of the homeostatic machinery and shows that cell survival threshold is not static but may be improved by previous environmental conditioning, following the principles of *adaptive homeostasis* (*Davies, 2016*). A unique feature of our study is to provide a framework to quantitatively explain the origin of this anticipatory behavior, as well as to link this phenomenon to the nonlinearity of stress doses responses (*Zhang and Andersen, 2007*; *Zhang et al., 2010*). Previous work has pointed to the existence of cross-protective effects, whereby exposure to a particular stress (e.g. heat, ethanol) increases tolerance to subsequent oxidative stress (*Mitchell et al., 2009*). Since Prx genes are also regulated by the transcription factors of the general stress response, Msn2/Msn4, we envision that the induction of tolerance to $H_2O_2$ by other stressors may be mediated by the same principles as those described here. More generally, it will be interesting to see how the proposed mechanism of tolerance applies to other homeostatic systems in which nonlinear stressor degradation might be conserved (*Zhang and Andersen, 2007*).

The last facet of nonlinear feedback regulation that emerges from our quantitative study is related to the concept of hormesis (*Calabrese et al., 2015*). Due to the substantial upregulation of Prxs in response to very low doses of $H_2O_2$, the tradeoff between beneficial and deleterious effects of $H_2O_2$ leads to a biphasic effect on cellular longevity as a function of $H_2O_2$ level. The effect of $H_2O_2$ on longevity provides a striking model of lifespan extension induced by environmental perturbation, similar to the effects of caloric restriction. Interestingly, this observation not only complements the previously described hormesis-like effect of $H_2O_2$ on chronological aging (*Mesquita et al., 2010*) but also suggests a simple mechanistic explanation in which the $H_2O_2$ homeostatic machinery plays a central role. Therefore, our study emphasizes that complex functional

properties, such as acquired stress tolerance and hormetic lifespan effects, can only be understood by developing specific approaches that bridge the molecular architecture of regulatory pathways to its quantitative dynamical properties.

## Materials and methods

### Strains and plasmids

All strains were congenic to S288C (*Sikorski and Hieter, 1989*; *Huh et al., 2003*) unless specified. Strains generated in this study were made using standard genetic techniques or classical PCR-mediated genome editing. See *Supplementary file 1* for the list of strains used in this study.

The transcriptional reporter strains *TSA1pr-sfGFP-deg*, *TRX2p-sfGFP-deg* were generated by a one-step cloning-free method (*Huber et al., 2014*). The promoters of *TSA1*, *TRX2* were duplicated in strain SY992 and a sfGFP-degron (superfolder GFP fused to CLN2 PEST sequence) tag was simultaneously inserted downstream of them. The genes *TSA1*, *TRX2* remain fully intact. The sequences of homology (used to design the primers) of the PCR products that duplicated the promoters at their genomic locations *in vivo* are listed below:

Sequences of homology used in PCR primers for promoter duplication in vivo

| Target locus | Sequence of homology to the target locus |
| --- | --- |
| TSA1pr-1 | TTCCCCTCGTTCAATTGCTCACAACCAACCACAACTACATACACATACATACACA |
| TSA1pr-2 | CCTTGATCTGGCTAAACTTGACTTCGTCAATTTCATTCAAGTGGAGATAGTCTCG |
| TRX2pr-1 | TTATACACGCACACATACACGAGAGTCTACGATATCTTTAAATAACACATCAATA |
| TRX2pr-2 | TGGATCATGGGCGCATGTGAACCGTACCCACCGAATTGCGCTTGAAGTGTGTCCA |

The strain *tsa1Δ; tsa2Δ; ACT1pr::TSA1-GFP* was generated by substituting the promoter region of *TSA1* by the promoter of *ACT1* in strain *TSA1-GFP (BY4741)*. The cassette kanMX4::ACT1pr was amplified by PCR and then transformed into the strain *TSA1-GFP (BY4741)* using standard lithium acetate protocol. The primers contained sequences homologous to the *TSA1* genomic locus that are listed below:

Sequences of homology used in PCR primers for promoter substitution

| Target locus | Sequence of homology to the target locus |
| --- | --- |
| TSA1-1 | CCACCGCCACAGGTGCGCAACCTCATCTCTACATTCCTGATGAAGACTAA |
| TSA1-2 | ACGGCAGTTTTCTTAAAAGTTGGAGCTTGCTTTTGAACTTGAGCGACCAT |

To introduce *tsa2Δ* mutation, the transformed strain was crossed with strain Y14287.

### H$_2$O$_2$ stability and dosage in the microfluidic setup

The H$_2$O$_2$ (Hydrogen peroxide solution 35wt. % in H$_2$O, 349887–500 ML, Sigma) was mixed in the SCD media at the suitable concentration prior the microfluidic experiment. In order to increase the stability of the H$_2$O$_2$ during the microfluidic experiments, the SCD media were kept on ice. Temperature measurements showed that this procedure does not affect media temperature in the microfluidic device since media reached room temperature before entering the device (data not shown). The H$_2$O$_2$ concentration was measured using a colometric H$_2$O$_2$ assay kit (OxiSelect Hydrogen Peroxide Assay Kit (Colorimetric), STA-343, EUROMEDEX, France) in media samples taken from the outlet of the chip.

Dosage experiments were run in triplicates.

### Consumption of H$_2$O$_2$ in bulk experiment

Over-night SCD-media culture (strain WT BY4742) was diluted in SCD to OD600 = 0.1 and incubated at 30°C, 220 rpm. At OD600 = 0.5, H$_2$O$_2$ was added to final concentration of 0.4 mM. Consumption

of the $H_2O_2$ was determined by measuring the $H_2O_2$ concentration in the cell media over time. Bulk $H_2O_2$ consumption experiments were performed six times.

## Microfabrication and microfluidics setup

Microfluidic chips were designed and made using standard techniques as previously described (*Fehrmann et al., 2013*). The microfluidic master used to assess the adaptation to oxidative stress was made using standard SU-8 lithography process at the ST-NANO facility of the IPCMS (Strasbourg, France). The microfluidic master for aging studies was made using similar techniques in the FEMTO-ST nanotechnology platform of the French Renatech network (Besançon, France). Prototypic molds were replicated in epoxy to ensure long-term preservation. The micro-channels were cast by curing PDMS (Sylgard 184,10:1 mixing ratio) and then covalently bound to a 24 × 50 mm coverslip using plasma surface activation (Diener, Germany).

Microfluidic chips were connected using Tygon tubing and media flows were driven by a peristaltic pump (Ismatec, Switzerland) with a 30 µL/min flow rate. Media switches -synthetic complete 2% dextrose (SCD), with or without $H_2O_2$ at the appropriate concentration- were performed using a computer-controlled electro-valve (Biochemfluidics) or using a media multiplexer (Elvesys). Linear ramps of medium containing $H_2O_2$ were achieved using a custom setup allowing a progressive increase in $H_2O_2$ concentration in the main medium tank.

For aging experiments, cells were maintained in a chip during typically 140 hr with constant medium perfusion (flow rate 10 µL/min). Cells started to enter the cavity about 20 hr following their loading in the device. In case $H_2O_2$ was used throughout the assay, fresh medium was prepared every 24 hr to prevent any decay in $H_2O_2$ concentration over time.

## Media diffusion through the microfluidics device/ effects of cell confinement

The diffusion of the media in the trapping cavities was tested using a fluorescein dye assay. To do this, fluorescein (Sigma) was flown trough the microfluidics device either in the absence or in the presence of a dense microcolony of cells and fluorescence images were snapped every 2 s. The fluorescence was measured both in the supply channel at the border of the trapping cavity and in the middle of the trapping cavity. As expected, the fluorescence in the supply channel increased almost instantaneously as shown in *Figure 1—figure supplement 1*, whereas the increase of the fluorescence in the middle of the empty trapping cavity displayed a delay (half rising time $t_{1/2}$ = 21 s). A comparable time ($t_{1/2}$ = 24 s) was obtained in the presence of a dense microcolony of cells, hence showing that diffusion is not impaired in a crowded environment.

## Time-lapse microscopy

All time-lapse experiments have been performed at least two times.

Freshly thawed cells were grown overnight at various final cell densities. In the morning, log phase cells were allowed to grow a few divisions and were transferred into the microfluidic device. Cells were imaged using an inverted Zeiss Axio Observer Z1 (adaptation assay) or a Nikon Tie (aging experiments). Focus was maintained using dedicated hardware throughout the assays. Fluorescence illumination was achieved using LED light (precisExcite, CoolLed or Lumencor) and light was collected using a 100× N.A. 1.4 objective and an EM-CCD Luca-R camera (Andor; adaptation experiments) or an Hamamatsu Orca Flash 4.0 (Aging experiments).

We used automated stages in order to follow up to 20 (adaptation experiments) or 60 (aging experiments) positions in parallel over the course of the experiment. Images were acquired every 3 min (adaptation experiments) or 10 min (aging experiments).

Temperature control was achieved using custom sample holder with thermoelectric modules and an objective heater with heating resistors. Temperature control was achieved using a PID controller (5C7-195, Oven Industries).

## Image analysis

Raw images were processed using custom software, called phyloCell, based on MATLAB and the image-processing toolbox (*Fehrmann et al., 2013*; *Paoletti et al., 2016*). This software features a comprehensive graphical user interface to perform segmentation/tracking and to introduce manual

error corrections. The software is available for download on GitHub (*Charvin, 2017*). A copy is archived on https://github.com/elifesciences-publications/phyloCell. In this study, the software was used to segment cell contours based on phase-contrast images; to track cells over time; and to measure the fluorescence within the cells, including nuclear localization of fusion proteins.

## Quantification of growth rate

After segmentation of cell contours from time-lapse data, the volume V of individual cells was estimated by spherical approximation of the cellular shape (*Figure 1—figure supplement 2A*). The volume increase rate (which we refer to as cellular growth rate) was then determined for each time point as the increase in the cell volume in consecutive frames per unit of time (cell growth rate at t1 = (V(t2)-V(t1))/(t2-t1), where t2-t1 = 3 min). More specifically, the measurements of mean growth rate per cell used throughout this study (which reflects the metabolic capacity of a population of cells, see below) were measured by averaging the volume increase rate of individual *budded* cells over large micro-colonies of cells, whereas *unbudded* cells were not considered in this analysis. This is motivated by two reasons: first, the increase in cell volume is higher during the budded phase of the cell cycle, and therefore, it provides a more robust estimate of cell growth. Second, the purpose of this measurement is to provide a readout of cellular metabolic activity, independently of cell cycle progression. When cells are exposed to $H_2O_2$, the fraction of budded cells increases due to the activation in some cells of a G2/M checkpoint, but this cell cycle arrest is not necessarily accompanied by a change in metabolic activity. Therefore, if we were pooling the growth rate of both unbudded (slow volume increase) and budded cells (fast volume increase), the change in the repartition of each type of cells upon $H_2O_2$ exposure would induce a misleading change in the mean growth rate per cell, which would be unrelated to a potential change in metabolic activity. Instead, selecting only the budded cells provides a relevant assessment of cell growth rate, which is unaffected by the inherent variations in cell cycle progression during stress exposure (*Figure 1—figure supplement 2B*).

Noteworthy, the definition of growth rate used in our study (dV/dt) differs from the classical one, which is given by $\mu = (1/V) \times dV/dt$. The reason why we used the simple first time derivative of cell volume is because the exponential growth model appeared to be irrelevant to assess the change in metabolic activity during stress exposure. Indeed, the growth rate of cells (defined as dV/dt) appeared to be independent of cell volume (*Figure 1—figure supplement 2C*). In addition, using dV/dt as readout of growth rate, we observed a complete recovery following the exposure to stress, suggesting that cells had recovered a normal metabolic activity. However, using $\mu = (1/V) \times dV/dt$ as the definition of growth rate, we would have missed the recovery in metabolic activity, since the mean cell volume of adapted cells following the recovery was higher than that before the stress (*Figure 1—figure supplement 2D*), due to cell cycle arrested – yet metabolically active-cells.

## Quantification of Yap1 nuclear localization

Following the segmentation of the nucleus using the Htb2-mCherry signal (*Figure 1D*), nuclear Yap1-GFP localization was measured at each frame by subtracting the average cytoplasmic fluorescence level from the mean nuclear level in order to remove the background. In addition, we noticed that there was a drift in the fluorescence level over time, which, for unclear reasons, appeared to mostly affect the cytoplasmic level. Therefore, we used a no-stress control experiment, in which Yap1-GFP nuclear localization was supposedly constant, to measure the extent by which background subtraction modifies the quantification of the nuclear localization over time. This measurement was then used to correct the drift observed in experiments in which a stress was applied.

## Quantification of cell survival using a propidium iodide (PI) assay

Media containing 5 μg/ml PI (Sigma, Saint-Louis, MO, USA) was flown through the microfluidics device during a 0.6 mM $H_2O_2$ step experiment, and images were recorded continuously. Dead cells, which experienced a loss of cell membrane integrity, incorporated the dye and displayed red fluorescence, as reported in *Figure 2—figure supplement 1*.

## Quantification of the phase-contrast brightness of the cells

We used the presence of persistent bright vacuoles as a readout to characterize the cells experiencing a permanent growth arrest following a $H_2O_2$ stress. To quantify this, we measured the mean intensity of the highest decile in the phase-contrast channel of the images.

## Protocol to generate linear stress ramps

In order to generate ramps of $H_2O_2$ stress, we used an extra peristaltic pump driving the flow of a $H_2O_2$ stock solution to gradually increase the $H_2O_2$ concentration in the medium tank used to feed the microfluidic device (*Figure 3—figure supplement 2A*). Assuming $C_1$ and $V_1$, the concentrations and volume of the medium tank, respectively, $\mu_1$ the flow rate used to perfuse the microfluidic device from the tank, $C_0$ and $\mu_0$ the concentration and flow rate from the $H_2O_2$ stock solution, using the conservation of mass, one can derive the evolution of $V_1$ and $C_1$ with time:

$$\frac{dV_1}{dt} = \mu_0 - \mu_1$$

$$\frac{dC_1}{dt} = \frac{\mu_0}{V_1^0 + (\mu_0 - \mu_1)t} (C_0 - C_1)$$

In the particular case of identical flow rates ($\mu_0 = \mu_1$), $V_1$ is constant and the evolution of $C_1$ with time is given by:

$$C_1 = C_0 \left( 1 - e^{\delta t} \right)$$

where $\delta = \frac{\mu_0}{V_1^0}$. When t << 1/δ, then the concentration $C_1$ increases linearly as:

$$C_1 = C_0 \, \delta \, t$$

With μ0 = 30 μL/min and $V_1^0$ = 1L, δ = 3. $10^{-5}$ $min^{-1}$. Therefore, we should get a linear slope of 3 μM/min using $C_0$ = 100 mM, provided t << $10^5$ min, which is much higher than the duration of experiments (typically $10^3$ min).

However, $H_2O_2$ dosages during the calibration of stress ramp experiments showed that the actual concentrations were systematically 21.5 ± 2% lower than expected for all tested time points (0–1000 min) and ramp slopes (1.4–22.4 μM/min) (data not shown). To explain the origin of this deviation from the expected measurements (which stands higher than the 10% decay observed during step experiments, see *Figure 1C*), we hypothesize that it can be attributed to a higher $H_2O_2$ degradation rate in the medium tank during ramp experiments because the tank was not kept on ice due to the need to perform constant mixing using a magnetic stirrer. Therefore, the actual evolution of concentration in the medium tank is given by:

$$C_1 = 0.785 \, C_0 \, \delta \, t$$

For instance, a 1.1 μM/min ramp slope is obtained with μ0 = 30 μL/min and $V_1^0$ = 1L and $C_0$ = 46.67 mM.

## Mathematical model of the $H_2O_2$ homeostatic machinery

To understand the mechanism that underlies $H_2O_2$ homeostasis, we developed a mathematical framework based on the negative feedback regulation in the Yap1 network. The model was used throughout this study to help identify and formalize the emergent properties of this system through iterative cycles of predictions and experimental challenges, rather than to perform exhaustive data fitting to retrieve individual parameter values. In the following, we describe in details the principles of the mathematical model.

### A - Model assumptions (refers to *Figure 3*)

The model describes the dynamics of scavenging of internal $H_2O_2$ (variable H) by antioxidants enzymes. Yap1 activates the synthesis of mRNAs from antioxidant genes ($A_{mRNA}$), which are in turn

translated into antioxidant enzymes (variable A) in response to an input $H_2O_2$ stress (parameter I). The evolution of $A_{mRNA}$, A and H with time can be written as:

$$\frac{dH}{dt} = \varepsilon + \alpha\,(I - H) - \beta\,A\,f(H) \tag{1}$$

$$\frac{dA_{mRNA}}{dt} = \gamma\,g(H) - \mu\,A_{mRNA} \tag{2}$$

$$\frac{dA}{dt} = \gamma'\,A_{mRNA} - \mu'\,A \tag{3}$$

where $\varepsilon$ represents the internal $H_2O_2$ production rate due to cellular activity, $\alpha$ is the rate constant associated with $H_2O_2$ diffusion, $\beta$ is the rate of $H_2O_2$ scavenging by antioxidants, $\gamma$ and $\gamma'$ are the transcription and translation antioxidants, respectively, and $\mu$ and $\mu'$ are the degradation rate constants associated with mRNA degradation and antioxidant dilution (due to cell growth), respectively; $f$ and $g$ are functions of H that characterize the sensitivity to $H_2O_2$ of scavenging and A transcription, respectively.

Although the purpose of this modeling approach is to make the most simple mathematical description using a limited set of variable in order to get an in-depth understanding of the distinct functional role played by each model feature, we chose to describe antioxidant transcription and consider mRNA level as a separate variable because it introduces a delay in the antioxidant response which is mandatory to explain experimental results reported in *Figure 3* (see also *Figure 3—figure supplement 1*).

For the sake of simplicity, the effect of $H_2O_2$ on cellular physiology (i.e. cell growth, overall transcription and translation machinery) is not considered in the present model, even though it is likely to provide additional regulatory feedbacks and modify the detailed dynamics of activation of the homeostatic system. In particular, we deliberately chose to consider the cellular growth rate μ'as a constant, even though our experiments clearly indicated that growth rate is transiently affected during step experiments. This assumption, which does not alter the generality of our analysis, is justified below.

Last, even though this model features an important number of parameters (i.e. 7), the analysis below reveals that the detailed dynamical properties of the system are only dependent upon a combination of parameters, but not on the individual values.

## B- Steady state for the linear model: *f(H)=1, g(H) = H* (refers to *Figure 1F* and *Figure 3*)

In this section, we assume that the $H_2O_2$ scavenging rate depends on antioxidant concentration A but is independent of internal $H_2O_2$ level H, that is $f = 1$. In addition, we hypothesize that the antioxidant transcription rate linearly increases with H.

Under these assumptions, it can be shown that the equilibrium state of the system described in *Equations (1-3)* is such that:

$$H_{eq} = \frac{\mu\mu'(\varepsilon + \alpha I)}{\alpha\mu\mu' + \beta\gamma\gamma'} \tag{4}$$

$$A_{mRNA,eq} = \frac{\gamma\mu'(\varepsilon + \alpha I)}{\alpha\mu\mu' + \beta\gamma\gamma'} \tag{5}$$

$$A_{eq} = \frac{\gamma\gamma'(\varepsilon + \alpha I)}{\alpha\mu\mu' + \beta\gamma\gamma'} \tag{6}$$

In the following, we set $\varepsilon = 0$ for the sake of simplicity, unless specified otherwise.

## C- Particular case: the 'integral feedback' model: µ'=0 (refers to *Figure 1F*)

From *Equation (4)*, in the particular case in which µ'=0 (i.e. antioxidant dilution can be neglected), the equilibrium is such that the internal $H_2O_2$ concentration is strictly zero: $H_{eq} = 0$, independently of the input H2O2 level I. This indicates that the system displays perfect homeostatic properties. This case corresponds to an 'integral feedback' system (*Muzzey et al., 2009*; *Yi et al., 2000*), in which the response of the scavenging machinery (production of antioxidants) is proportional to the integral over time of a function F (which can be derived from *Equations (1) and (2)*) that only depends on H, as follows (2):

$$A = \gamma \int F(H)dt \tag{7}$$

Therefore, during the transient regime following the exposure to a step in $H_2O_2$, as long as $F(H)$ is not strictly zero, A keeps building up, thus ensuring a complete scavenging of internal $H_2O_2$ (i.e. $H_{eq} = 0$). *Figure 3—figure supplement 1A* illustrates this property of the model, by showing the evolution of model variables upon exposure to a given $H_2O_2$ concentration. Such integral feedback control is widely used in systems engineering to ensure perfect tuning of a controlled variable (e.g. temperature control).

However, the integral feedback model is ruled out by the experimental observation that $H_{eq}$ is not zero and depends on the external $H_2O_2$ level (*Figure 1F*). In contrast, with µ'≠0, $H_{eq}$ is proportional to the external $H_2O_2$ concentration (*Equation (4)*), in agreement with *Figure 1F*.

Therefore, the enzyme dilution rate (set by µ') directly controls internal stress concentration at steady-state and therefore the accuracy of the homeostatic system. However, numerical simulations in *Figure 3—figure supplement 1A and B* show that it does not impact the transient increase in internal $H_2O_2$ levels during a step experiment, presumably because other time constant governs the kinetics of activation of the negative feedback loop. Therefore, the simulations in *Figure 3—figure supplement 1A and B* demonstrate that a temporary variable growth rate (as experimentally observed), would not change the transient dynamics of the Yap1 signaling pathway and therefore would have no effect on adaptation to $H_2O_2$ steps.

In addition, taking this observation into consideration would require to model the additional feedback of internal $H_2O_2$ level on enzyme dilution rate. This additional feature would introduce the following drawbacks: (1) the model would no longer be linear (since the enzyme dilution rate would necessarily be a nonlinear function of internal $H_2O_2$ and antioxidants concentration); (2) by making the model more cumbersome, it would be difficult to interpret the respective effects of each feature of the model. Therefore, for sake of simplicity, this possibility has not been investigated in the scope of this study.

## D- Response to linear ramps for the linear model (refers to *Figure 3*)

In this section, we calculate the response of the model to a linear ramp of $H_2O_2$, defined by:
$I(t) = \delta\, t$, where δ is the slope of the ramp (µM/min) and t is the time.

### Particular case of the integral feedback (µ'=0)

Even though this case does not match experimental findings (see above), it is instrumental to calculate the behavior of this system when it is submitted to linear ramps.

Under this assumption, we can show that A follows:

$$A(t) \approx \frac{\alpha\delta t}{\beta} \tag{8}$$

$A_{mRNA}$ is given by:

$$A_{mRNA}(t) = \frac{1}{\gamma'}\frac{dA}{dt} \approx \frac{\alpha\delta}{\beta\gamma'} \tag{9}$$

and H is given by:

$$H(t) = \frac{\mu}{\gamma} A_{mRNA} \approx \frac{\alpha\mu\delta}{\beta\gamma\gamma'} \tag{10}$$

Therefore, interestingly, the internal $H_2O_2$ does not vary with time and its magnitude is proportional to the slope of the ramp $\delta$. More specifically, since cells should adapt as long as $H(t)_{tox}$, this predicts that cells can tolerate a maximum slope $\delta_{max}$ defined by:

$$\delta_{max} = H_{tox} \frac{\beta\gamma\gamma'}{\alpha\mu}$$

(11)

*Figure 3—figure supplement 1C* illustrates the plateau of H reached during linear ramps for the integral feedback model.

### General case of the linear feedback model (μ'≠0)

In this case, unlike the integral feedback model, it can be shown that H increases linearly with time as:

$$H \approx \frac{\mu\mu'\alpha\delta t}{\alpha\mu\mu' + \beta\gamma\gamma'}$$

(12)

as shown on the numerical simulation on *Figure 3—figure supplement 1D*. Therefore, under this assumption, the system has weaker homeostatic capabilities, compared to the integral feedback case.

Unlike the transient regime observed in step experiments, under ramping stress, the evolution of H with time greatly depends on the dilution rate μ'. Therefore, in this case, it is probable that the growth rate arrest observed with steep ramps may help the cell survive to higher absolute stress levels by allowing more time for the cells to accumulate antioxidants enzymes, provided that their synthesis rate is not affected by stress. This possibility can be better understood by considering the adaptation to an arbitrary stress pattern, using the 'phase diagram' described in the next section. On this plot, the iso-H line on the phase diagram delimitating cell survival and arrest would be in between those displayed on *Figure 3—figure supplement 1E* (μ'=0) and *Figure 3—figure supplement 1F* (μ' ≠ 0).

### E- Characterization of the training capabilities – 'phase diagram' (refers to *Figure 3*)

According to the analysis performed in the previous section, the internal $H_2O_2$ level reached upon stress exposure is proportional to the magnitude of the stress I during step experiment, but also strongly depends on the slope d = dI/dt of the ramp when using linearly increasing stress. By combining these two types of external perturbations, we can calculate numerically the maximal internal $H_2O_2$ level $H_{max}$ reached when the system is submitted to a stress of magnitude I applied with a rate $\delta$ (as described on the two-dimensionnal plot in *Figure 3C*, *Figure 3—figure supplement 1E and F*).

We assume that there is toxicity threshold $H_{tox}$ (see *Figure 3C*) that is defined by the transient internal $H_2O_2$ level obtained when the cells are exposed to the limit concentration of 0.6 mM $H_2O_2$ in step experiments. Based on this, the 'iso-H' line defined by $H_{max} = H_{tox}$ splits the (I, d = dI/dt) space into a domain where adaptation is possible ($H_{max} < H_{tox}$) and a domain where adaptation is prohibited ($H_{max} > H_{tox}$). For the linear models, the iso-H lines can be calculated analytically, see below. The interest of such 'phase diagram' is that it provides a synthetic overview the homeostatic capabilities of the system in response to an arbitrary linear perturbation.

### Particular case of the integral feedback (μ'=0)

In this particular case, in addition to the 0.6 mM threshold for step experiments (see vertical dashed line on *Figure 3—figure supplement 1E*), there is a threshold ramp slope $\delta_{max}$ (defined in the previous section) beyond which adaptation is prohibited (horizontal line on *Figure 3—figure supplement 1E*). Conversely, this model implies that there is no limit in adaptation provided the slope of the ramp is less than $\delta_{max}$. In practice, however, this prediction is challenged by experimental observation (*Figure 3*) showing that there is a limited range of I allowing cell adaptation, even with very low ramp slopes. Therefore, the experiments focused on adaptation to a linear ramp further invalidate the integral feedback model.

## General case of the linear model (μ'≠0)

In this case, in addition to the 0.6 mM threshold for step experiments (see vertical dashed line on *Figure 3—figure supplement 1F*), one can calculate the asymptotic limit of the iso-H line ($H_{max}$ = $H_{tox}$) for slow ramps:

$$\delta = \mu'(I_{abs} - I) \tag{13}$$

where:

$$I_{abs} = H_{tox}\left(1 + \frac{\beta\gamma\gamma'}{\alpha\mu\mu'}\right) \tag{14}$$

*Equation (13)* has two important implications: first, the maximum allowed ramp slope $\delta$ is a linear function of the magnitude of the stress I, and its slope is set only by the antioxidant dilution rate $\mu'$. Second, there is an absolute H₂O₂ concentration $I_{abs}$ beyond which no adaptation is possible, as described below (*Figure 3—figure supplement 1F*).

## Absolute adaptation concentration I_abs

*Equation (14)* defines the absolute H₂O₂ concentration $I_{abs}$ that allows cellular adaptation. As expected intuitively, it appears as the balance between the kinetic constants that favor adaptation (scavenging rate, gene expression rate) and the ones that tend to increase internal H₂O₂ levels (H₂O₂ diffusion rate, dilution rate of antioxidant mRNA and proteins). Therefore, the experimental measurement of $I_{abs}$ (as shown in *Figure 3—figure supplement 2F*) provides an assessment of the overall homeostatic capabilities of the system.

## Influence of the delay in antioxidant response on the training capabilities

In addition to experimental measurement of growth rate during ramp experiments, *Figure 3F* displays the iso-H line ($H_{max}$ = $H_{tox}$) as expected by the model, in which parameters values are similar to *Figure 3—figure supplement 1I*. The set of parameters was chosen to match the experimentally determined $I_{abs}$ = 7.3 mM. Interestingly, in these conditions, the model tends to overestimate the adaptation capabilities for steep ramps ($\delta$ > 10 μM/min). We interpret this disagreement by the fact that there is a delay in the synthesis of antioxidants that render the system more sensitive to steep ramps. Indeed, by changing the kinetics of mRNA stability (by adjusting the value of decay rate to μ = log(2)/40 min$^{-1}$ [*Geisberg et al., 2014*]), while maintaining the same value for $I_{abs}$, we obtained a better agreement with experiments (see *Figure 3F* and *Figure 3—figure supplement 1G–1J*).

## F- Steady state for the nonlinear model: *f(H) = H/(H+W), g(H) = H/(H+K)* (refers to *Figures 1* and *6*)

This model constitutes a refinement of the linear feedback model described above, which fails to describe the scaling of Tsa1-GFP level with increasing H₂O₂ levels (*Figure 6*).

### Saturation in antioxidant transcription rate

To this end, the first assumption to be added in the model is linked to the observed saturation in Tsa1 transcription rate (*Figure 6—figure supplement 1C*). Assuming g(H) = H/(H+K), where K is a constant that sets the saturating H concentration, the steady-state level for H is given by:

$$H_{eq} = \frac{1}{2}\left(-(Z + K - I) + \sqrt{(Z + K - I)^2 + 4IK}\right) \tag{15}$$

with: $Z \equiv \frac{\gamma\gamma'\beta}{\mu\mu'\alpha} \sim I_{abs}$

Therefore, even though $H_{eq}$ is *a priori* a nonlinear function of I, when I << $I_{abs}$, $H_{eq}$ varies linearly with I:

$$H_{eq} \approx I\frac{K}{Z} \tag{16}$$

Similarly, $A_{eq}$ is given by:

$$A_{eq} \approx I \frac{\alpha}{\beta} \left( 1 - \frac{K}{Z} \right)$$

In contrast, when $I = I_{abs}$ and above, the equilibrium is given by:

$$A_{eq} \approx \frac{\gamma\gamma'}{\mu\mu'} \tag{17}$$

and

$$H_{eq} \approx I - Z \tag{18}$$

which is expected since, under these conditions, the system has reached its maximum scavenging capabilities.

Therefore, over an extensive range of concentration (since $I_{abs}$ was shown to be larger than 7 mM, see *Figure 3—figure supplement 2F*), the level of antioxidants should scale linearly with the stress level I. This prediction is contradicted by the nonlinear scaling observed experimentally, especially for low $H_2O_2$ values (*Figure 6*), therefore indicating that an additional assumption is required to explain the data.

### Nonlinear $H_2O_2$-dependent scavenging rate

In the following, we assume that, in addition to saturation in antioxidant transcription rate, the rate of $H_2O_2$ scavenging by antioxidants is set by: f(H) = H/(H+W), where W represents the internal $H_2O_2$ concentration at which the scavenging rate saturates. In addition, unlike the previous sections, we now hypothesize the existence of a non-zero basal internal $H_2O_2$ production rate ε.

The steady-state levels $H_{eq}$ and $A_{eq}$ are given by the solution of a third-degree equation. However, in the particular case in which the steady state is below the saturation level for scavenging and transcription rate (i.e. $H_{eq} \ll K,W$, corresponding to low external $H_2O_2$ levels), $H_{eq}$ can be approximated as:

$$H_{eq} = \frac{1}{2} \left( -\sigma + \sqrt{\sigma^2 + 4\sigma\left(I + \frac{\varepsilon}{\alpha}\right)} \right) \tag{19}$$

with $\sigma = \frac{\alpha\mu\mu'KW}{\beta\gamma\gamma'} \sim \frac{KW}{I_{abs}}$, and:

$$A_{eq} = \frac{\gamma\gamma'}{\mu\mu'} \frac{H_{eq}}{K} \tag{20}$$

Therefore, this models implies that there is a sublinear dependency of both A and H with increasing stress level I, even for low external $H_2O_2$ levels, as indeed observed on *Figures 1F* and *6C*. Noteworthy, the magnitude of the nonlinearity is controlled by three parameters: σ, which combines the saturation concentrations K and W, and the ratio ε/α, which represents the internal $H_2O_2$ concentration in the absence of scavenging and external $H_2O_2$.

## G- Acquisition of tolerance

In our experiments (*Figure 4*), cells that have been pre-exposed to $H_2O_2$ display a lower burst of nuclear relocation of Yap1-GFP upon a mild subsequent challenging stress, compared to naive ones.

A numerical integration of *Equations (1-3)* according to the hypotheses of linear versus nonlinear feedback models is displayed in *Figure 4—figure supplement 1E and F*, using the same set of parameters as in *Figure 6*.

## H- Summary of parameters values, data fitting procedures and numerical integration

### Linear model

Parameters used in the linear model

| Parameter name | Value | Comment |
|---|---|---|
| ε | 0 or 0.2 mM min$^{-1}$ | Basal $H_2O_2$ production rate in the absence of external $H_2O_2$. In the integral feedback case (μ'=0), the parameter value must be 0, otherwise there is no stable steady-state. |
| α | 1 min$^{-1}$ | $H_2O_2$ transport rate across cell membrane.<br>Value based on the order of magnitude of timings (~1 min) of nuclear relocation upon addition of $H_2O_2$ (taken from *Figure 1—figure supplement 1C*). |
| β | 5 min$^{-1}$ | $H_2O_2$ scavenging rate. Derived from the absolute adaptation concentration $I_{abs}$ in *Figure 3*. |
| γ | 0.03 or 0.53 min$^{-1}$ | Antioxidant transcription rate. This parameter was adjusted to keep a constant value of $I_{abs}$ when changing the mRNA decay rate (see below). |
| μ | variable | Antioxidant mRNA decay rate.<br>In *Figure 3B* we used: μ=log(2)/2.5 min$^{-1}$ to display the general properties of the dynamics of the system.<br>In *Figure 3C,F*, we used: μ=log(2)/40 min$^{-1}$, which provides a better agreement with experimental data, and corresponds to the actual order of magnitude of mRNA stability for TSA1 and other Prx genes(*Geisberg et al., 2014*). |
| γ' | 0.01 min$^{-1}$ | Antioxidant translation rate. Derived from the absolute adaptation concentration $I_{abs}$ in *Figure 3*. |
| μ' | log(2)/100 min$^{-1}$ | Antioxidant protein decay rate. Tsa1 is a stable protein, therefore, the decay is due to the dilution rate set by cell growth (doubling time ~ 100 min). |

## Data fitting procedure

Values for α, μ and μ' are set by experimental constraints as described in the table above. γ, γ' and β could not be determined individually using the experiments performed in this study, since absolute concentration of H, A and $A_{mRNA}$ could not be measured. However, these parameters are tightly linked to $I_{abs}$, which is the absolute $H_2O_2$ concentration at which cells can adapt (as described in section 1E in this document). Experimentally $I_{abs} = 7.2$ mM, as obtained in *Figure 3—figure supplement 2F*. Based on this measurement, putative values of γ, γ' and βwere set to get the expected value of $I_{abs}$.

## Nonlinear model

Parameters used in the nonlinear model

| Parameter name | Value | Comment |
|---|---|---|
| ε | 0.06 mM min$^{-1}$ | Basal $H_2O_2$ production rate in the absence of external $H_2O_2$<br>Determined by a fitting procedure to data in *Figure 6*. |
| α | 1 min$^{-1}$ | $H_2O_2$ transport rate across cell membrane. Taken from *Figure 1—figure supplement 1C*. |
| β | 10 mM$^{-1}$ min$^{-1}$ | $H_2O_2$ scavenging rate. Derived from the absolute adaptation concentration $I_{abs}$ in *Figure 3*. |
| γ | 0.1 mM$^{-1}$ min$^{-1}$ | Antioxidant transcription rate. Derived from the absolute adaptation concentration $I_{abs}$ in *Figure 3*. |
| μ | μ=log(2)/40 min$^{-1}$ | Antioxidant mRNA decay rate. See the linear model above for the determination of parameter value. |
| γ' | 0.005 mM$^{-1}$ min$^{-1}$ | Antioxidant translation rate. Derived from the absolute adaptation concentration $I_{abs}$ in *Figure 3*. |
| μ' | log(2)/100 min$^{-1}$ | Antioxidant protein decay rate. Tsa1 is a stable protein, therefore, the decay is due to the dilution rate set by cell growth (doubling time ~ 100 min). |
| K | 0.01 mM | Order of magnitude of internal $H_2O_2$ at which antioxidant. transcription saturates. Taken from *Figure 6—figure supplement 1C*. |
| W | 0.01 mM | Order of magnitude of internal $H_2O_2$ at which $H_2O_2$ scavenging rate saturates. The ratio K/W ~ 1 gave the best fit to curves in *Figure 6*, hence the order of magnitude of this parameter. |

## Data fitting procedure

In *Figure 6*, Tsa1-GFP level data fitting to the model (using chi2 minimization) allows to estimate the order of magnitude of ε, and the ratio K/W, assuming that other parameter values are similar to the linear model. Importantly, this parameter set is then used with no fit (other than a scaling factor due to the use of arbitrary units in experimental measurements) to compare the agreement between data and model in *Figure 1* (steps), *Figure 6* (ramps) and *Figure 4* (tolerance). Unlike the linear model, this model is unique in its ability to describe the acquisition of tolerance in pre-exposed cells.

## Numerical integration

Computing was performed using Matlab software. Numerical integration was based on the *ode45* and *ode15s* functions.

## Acknowledgements

We are grateful to Zhou Xu, Sophie Quintin, Nacho Molina, Robert Schneider and Julien Vermot for careful reading of the manuscript. This work was supported by the French Association pour la Recherche contre le Cancer (YG), the ATIP-Avenir program (GC), and a grant from the Fondation pour la Recherche Médicale (GC and MBT). This study was supported by the grant ANR-10-LABX-0030- INRT, a French State fund managed by the Agence Nationale de la Recherche under the frame program Investissements d'Avenir ANR-10- IDEX-0002-02.

## Additional information

### Funding

| Funder | Grant reference number | Author |
| --- | --- | --- |
| Association pour la Recherche sur le Cancer | PDF20111204470 | Youlian Goulev |
| Agence Nationale de la Recherche | ERRed | Michel B Toledano |
| Institut National Du Cancer | PLBIO INCA_5869 | Michel B Toledano |
| Institut National de la Santé et de la Recherche Médicale | ATIP-Avenir program | Gilles Charvin |
| Fondation pour la Recherche Médicale | DEI20151234397 | Gilles Charvin |
| Agence Nationale de la Recherche | ANR-10-LABX-0030-INRT (French State fund ANR-10-IDEX-0002-02) | Gilles Charvin |

The funders had no role in study design, data collection and interpretation, or the decision to submit the work for publication.

### Author contributions

YG, Conceptualization, Resources, Data curation, Software, Formal analysis, Investigation, Methodology, Project administration, Writing—review and editing; SM, Methodology, Designed the microfluidic device used for RLS analysis; AM, BH, MM, Resources, Designed the yeast strains used in this study; MBT, Conceptualization, Supervision, Writing—review and editing; GC, Conceptualization, Data curation, Software, Formal analysis, Supervision, Funding acquisition, Visualization, Methodology, Writing—original draft, Project administration

### Author ORCIDs

Youlian Goulev, http://orcid.org/0000-0003-0370-4567
Bo Huang, http://orcid.org/0000-0001-5945-7601
Gilles Charvin, http://orcid.org/0000-0002-6852-6952

## Additional files

### Supplementary files

• Supplementary file 1. List of all strains used in this study. Table listing the strains used in this study as well as additional information about the genotypes and the origins of the strains.

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
