## [Decision Letter]

Thank you for submitting your article "Nonlinear feedback drives homeostatic plasticity in H_2_O_2_ stress response" for consideration by *eLife*. Your article has been reviewed by two peer reviewers, and the evaluation has been overseen by Naama Barkai as the Senior Editor and Reviewing Editor. The reviewers have opted to remain anonymous.

The reviewers have discussed the reviews with one another and the Reviewing Editor has drafted this decision to help you prepare a revised submission.

In this study, the authors addressed quantitatively the adaptation of *S. cerevisiae* cells to oxidative stress. They used single cell readouts together with microfluidics and mathematical modeling to propose how cells acquire tolerance to H_2_O_2_. They found that the induction of peroxiredoxins by the Yap1 transcription factor (AP1) is a key phenomenon during adaptation in this particular yeast.

Both reviewers appreciated the high resolution of your microluidics experiments and the combination of mathematical modeling with quantitative experiments. However, the major concern, raised by reviewer #2 and agreed by the other reviewer, is that novelty is limited as the majority of biological findings are already known. This is a major concern which you need to address.

Despite this concern, the reviewers were positive about publication, as they greatly appreciated the depth of your measurements and analysis. A major concern here, though is the lack of cell-survival information, which the reviewers believe is the correct phenotype to be monitored when considering adaptation. As an essential revision, we therefore require that you will measure cell survival in your essays relevant for Figure 1–Figure 4.

*Reviewer #1:*

The authors study the dynamics of the response of *S. cerevisiae* to changing H_2_O_2_ levels. They use of microfluidics device that allow a precise temporal control of the media. A simple toy model that includes a negative feedback, of the stress response system is presented. The response of the growth rate, Yap1-GFP and TRX1pr-sfGFPdeg to a step response of H_2_O_2_ reveal a growth rate adaptation up to a critical H_2_O_2_ value together with an overshoot in Yap1-GFP that are consistent with a simple model termed 'linear'. These dynamics depend on the initial H_2_O_2_ the cells experience ('pre-treatment') and require the addition of Michaelian terms of the internal H_2_O_2_ levels (H) to the model (to the H_2_O_2_ scavenging term and the antioxidant production term). Mutant analysis shows that peroxiredoxin genes that were previously identified to be involved in the H_2_O_2_ stress response effect the adaptation to stress ramp. It is also shown that H_2_O_2_ levels have a non-monotonic effect on replicative life span that is Yap1 and Tsa1 dependent.

One of the main novelties of this study is the use of microfluidics that allows a temporal control of H_2_O_2_. It seems that the design of the chip relies on diffusion to carry nutrients from the feeding channel to the cells. This diffusion, in particular when cells are crowded in the channel where they grow has an effect on the effective H_2_O_2_ cells are experiencing and introduce a time delay that depends on the density of cells. The authors should provide arguments that these are negligible (either by adding a die and characterizing the flow or by simulation).

The authors divide the cells into 2 'types' of cells. The correlation between the type of cells and their location on the channel should be presented to rule out the cells in the middle of the channel are buffered.

One of the toy model assumptions is that the dilution time is constant. The authors mention in the Supplementary Information that although they realize that the growth rate is far from constant (which is actually one of the main results), it will not change the results. Taking into account the time dependence of the dilution time is critical to the understating of the underlying principles as advocated by the authors. The growth pause allows a significant time for protein to accumulate. The regulator overshot can rise from it. Without showing that the negative feedback that arise from the growth rate modulation is not significant (even by simulations and not analytical solution), the generality of the conclusion is questionable.

General

This study is elegant and the use of dynamical measurements and modeling is synergistic. Using negative feedback (and comparing it to integral feedback) and its features has been heavily studied (see for example, Khammash M. BMC Biology 2016). Feedback terms without saturation seldom describe the data well in biological systems. The reader would benefit from the use of the linear model as a pedagogical example to highlight the fact that overshoot ("training") can happen without dependence on initial conditions and hysteresis ("stress tolerance) also if it appears in the Supplementary Information. The paper would benefit from comparing the Michaelian model to a model where the growth rate is not constant in time. This will provide a vivid example for the interplay between growth rate and transnational feedback.

*Reviewer #2:*

Overall, the experiments presented here are quite well executed and the combination between experimental studies and mathematical modeling has been useful to validate some of their initial hypothesis. However, there are aspects of the study that limit its wide interest.

The information obtained from the study is mostly not novel, in contrast of what the authors claim. It has been known for oxidative stress and other stresses, in yeast and in other organisms, that 1) different kinetics of stress exposure leads to different adaptation patterns (training), specially on those systems in which transcription is involved and 2) that the pre-exposure to stress leads to acquisition of tolerance. It is also known that the Yap1 transcription factor is a major regulator of oxidative stress defense genes and that Yap1 regulates peroxiredoxin expression (major H_2_O_2_ scavengers). Thus, albeit nicely presented and quantitatively assessed, the new knowledge generated in the study is rather limited.

The experimental set up used to study adaptation is interesting and well executed. They mostly monitor Yap1 localization as a measure of oxidative stress signaling and growth rate to assess adaptation. These measurements are correct and they provide relevant information however, they fall short to conclude on cellular adaptation. Cell survival should be monitored (at least for Figure 1–Figure 4) to be able to claim on adaptation, measuring growth rate to visualize a transient cell cycle arrest is clearly not sufficient. Actually, cells delay cell cycle in response to stress and this delay is essential to maximize cell survival. The delay per se is an adaptive response and thus it cannot be concluded that cells that delay cell cycle are those that do not adapt properly.

Ramping experiments are interesting (Figure 3) to test the initial hypothesis. However, if I understood them properly, they do not reach the same H_2_O_2_ concentration. Authors should perform the same experiments increasing H_2_O_2_ at different rates but reaching the same final concentration (with different times) and then compare adaptation. Otherwise they are comparing not only different rate of exposure but also different final concentration of H_2_O_2_.

Negative feedback is proposed to explain some of the results. The authors do not study the contribution of such feedback in the training or acquired tolerance. This could be an interesting aspect to be analyzed.

---

## [Author Response]

*[…] Both reviewers appreciated the high resolution of your microluidics experiments and the combination of mathematical modeling with quantitative experiments. However, the major concern, raised by reviewer #2 and agreed by the second reviewer, is that novelty is limited as the majority of biological findings are already known. This is a major concern which you need to address.*

We thank the reviewers for the careful reading of our manuscript and the overall positive appreciation of our work. However, we obviously do not share their opinion regarding the lack of novelty. Here, we provide first a summary of the major experimental and conceptual findings, developed in detail in the point-by-point reply below.

Novel Conceptual Findings

The main novelty of our study is the development of a quantitative framework of cellular adaptation to stress, using the response to H_2_O_2_ as a model system. The combination of microfluidics with live cell fluorescence measurements provides an unprecedented assessment of the dynamics of activation of the Yap1 regulatory network and its consequences on cellular adaptation, which could not be performed using classical methodology. Most importantly, our experimental approach served as prerequisite to build a mathematical framework that captures emergent properties of the system, such as acquisition of tolerance and hormesis. These latter phenomena could not be successfully understood using previous qualitative approaches. Therefore, our study provides important conceptual advances in the field of stress response and physiological adaptation, which go beyond the classical homeostatic paradigm. We think that our approach, which establishes the Yap1 network as a quantitative model system for nonlinear negative feedback-based regulation, will be applied to other stress response contexts and, more generally, to other signaling pathways.

Novel experimental findings

1) A sharp H_2_O_2_ threshold

Classical H_2_O_2_ resistance colony formation assays reported in the literature were unable to capture the sharp concentration threshold observed in our study (related to Figure 1–Figure 2), nor the existence of cellular heterogeneity close to the survival threshold (related to Figure 2 and Figure 2—figure supplement 2, Figure 2—figure supplement 3). Our ability to quantitatively and temporally control the concentration the H_2_O_2_ was decisive for making these observations. In the revised manuscript, additional information regarding the fate of non-adapting cells has been included, which further refine this analysis (cell fate and survival analysis, see below).

2) Cellular adaptation to ramping H_2_O_2_ stress

The dynamical properties of a system based on negative feedback regulation have been extensively described in Control Theory. However, in the case of stress response, that cellular survival/adaptation might be stress-rate dependent has not been discussed formally (i.e. using a mathematical framework), with the exception of the theoretical model of the Hog1 system (Zi Z., …, Klipp E., PlosOne, 2010). In addition, to the best of our knowledge, this hypothesis has not been investigated experimentally. Last, the observation that cells can cope with a concentration of >7mM H_2_O_2_ is novel and reveals unprecedented adaptation capabilities of the H_2_O_2_ defense system. Adaptation at such high concentration could not have been observed without a stress ramping protocol, as performed in our study.

3) Mechanism of acquisition of tolerance to H_2_O_2_

As clearly stated in our Introduction, acquisition of tolerance is a long-standing phenomenon conserved across species from unicellular to mammals, for which a quantitative mechanistic understanding was missing. One of the main novelty of the paper is to propose a framework that explains the emergence of this property in the context of the H_2_O_2_ response. The novelty lies in the discovery of a missing mechanism for a long-known phenomenon.

4) Hormetic effect of H_2_O_2_ on cellular longevity

The biphasic dose-dependent effect of external H_2_O_2_ on replicative longevity (i.e. 27% increase at mild H_2_O_2_ levels) is totally novel. This observation is quite counter-intuitive at first sight, but can be explained by quantitative reasoning regarding the respective contributions of beneficial effects (nonlinear peroxiredoxin expression) and deleterious effects of H_2_O_2_ (DNA and protein damages).

While hormetic effects have been reported in other biological contexts, our discovery of a biphasic effect of external H_2_O_2_ level on longevity, along with the underlying molecular mechanism will be of major interest to the aging community. We shall stress the fact that this effect has probably been missed in previous studies, because maintaining a low (~ 25 µM) level of H_2_O_2_ is impossible in microdissection experiments, and therefore requires a constant media replenishment device, as in our microfluidic device.

In the revised version, we have edited the manuscript to emphasize the novelty of the experimental findings, as well as the originality of our approach.

Despite this concern, the reviewers were positive about publication, as they greatly appreciated the depth of your measurements and analysis. A major concern here, though is the lack of cell-survival information, which the reviewers believe is the correct phenotype to be monitored when considering adaptation. As an essential revision, we therefore require that you will measure cell survival in your essays relevant for Figure 1–Figure 4.

We appreciate the positive comments of the reviewers. We acknowledge that cell survival information was missing, and we thank reviewer #2 for pointing this out. The revised version includes an in-depth assessment of cell survival using standard techniques that alleviate potential ambiguities related to cellular adaptation upon exposure to hydrogen peroxide (see below for details).

*Reviewer #1:*

*[…] One of the main novelties of this study is the use of microfluidics that allows a temporal control of H_2_O_2_. It seems that the design of the chip relies on diffusion to carry nutrients from the feeding channel to the cells. This diffusion, in particular when cells are crowded in the channel where they grow has an effect on the effective H_2_O_2_ cells are experiencing and introduce a time delay that depends on the density of cells. The authors should provide arguments that these are negligible (either by adding a die and characterizing the flow or by simulation).*

We agree with the reviewer that this control experiment was missing in the original version of the manuscript. In the revised version, we have included a diffusion assay of a fluorescent dye (fluorescein) in the microfluidic device. This experiment, which was carried out in the presence or absence of a dense microcolony of cells, revealed that there is no significant delay associated with the presence of cells in the cavity. We have added a supplementary Figure 1—figure supplement 1 that describes these results.

*The authors divide the cells into 2 'types' of cells. The correlation between the type of cells and their location on the channel should be presented to rule out the cells in the middle of the channel are buffered.*

This is a totally fair concern that is linked to the previous one. To answer this question, we have retrieved the absolute location of the cells in the trapping cavity for stress resistance assays performed at sublethal stress concentration (0.5mM H_2_O_2_), at which a mixture of 3 different cellular phenotypes (adapted, prolonged cell cycle arrest, permanent growth arrest) is observed.

To maximize the statistical significance, we have split the cells into two groups, depending on their location in the cavity (either at the border, or at the center). By doing so, we have not found any bias associated with cell location regarding the distribution of the 3 phenotypes. We have added a panel in Figure 1—figure supplement 1 that describes this analysis, which was performed on existing data.

*One of the toy model assumptions is that the dilution time is constant. The authors mention in the Supplementary Information that although they realize that the growth rate is far from constant (which is actually one of the main results), it will not change the results. Taking into account the time dependence of the dilution time is critical to the understating of the underlying principles as advocated by the authors. The growth pause allows a significant time for protein to accumulate. The regulator overshot can rise from it. Without showing that the negative feedback that arise from the growth rate modulation is not significant (even by simulations and not analytical solution), the generality of the conclusion is questionable.*

We thank the reviewer for raising this important aspect of the model. Experimentally, it is obvious that the growth rate undergoes a transient decrease during the step experiments, and growth rate is indeed used as a readout of cell physiology throughout the paper.

In the theoretical part of the manuscript, we stress that enzyme dilution rate (due to cell growth) affects the behavior of the homeostatic system: in the absence of dilution (no growth), it behaves as a perfect integral feedback system (see Figure 3—figure supplement 1 and supplementary text). In contrast, assuming a non-zero dilution rate considerably restricts the adaptation limit during ramps (see Figure 3—figure supplement 1). The underlying reason is that dilution rate directly controls the *steady-state* level of both enzyme (A) and internal H_2_O_2_ concentration (H).

However, the dilution rate does not impact the *transient* increase in internal H_2_O_2_ levels during the simulation of a step experiment: see Figure 3—figure supplement 1 (no growth) versus Figure 3—figure supplement 1 (with growth). Similarly, the phase diagrams in Figure 3—figure supplement 1 (no growth) and Figure 3—figure supplement 1 (with growth) are identical when considering slopes steeper than 50µM/min. This is because the recovery time following a step is set by the activation rate of the negative feedback loop (enzyme synthesis and H_2_O_2_ scavenging) rather than by the restoring force due to the growth rate. Therefore, the identical profile of the internal stress dynamics in the simulations in Figure 3—figure supplement 1 (no growth) and Figure 3—figure supplement 1 (no dilution) demonstrate that a variable growth rate would not change the transient dynamics of the Yap1 signaling pathway and therefore would have no effect on adaptation to H_2_O_2_ steps.

In contrast to H_2_O_2_ steps, it is likely that the growth rate arrest observed in ramp experiments may help the cells survive to higher absolute stress levels by allowing more time for the cell to accumulate enzymes, as pointed by the reviewer. In this case, the iso-H line on the phase diagram delimitating cell survival and arrest would be in between those displayed on Figure 3—figure supplement 1 and Figure 3—figure supplement 1.

We thought that modelling the additional negative feedback of internal H_2_O_2_ level on enzyme dilution rate – even though it would make the model more accurate – would have the following drawbacks: 1) the model would no longer be linear (since the enzyme dilution rate would be a nonlinear function of internal H_2_O_2_ and antioxidants) 2) by making the model more cumbersome, it would be difficult to interpret the respective effects of each feature of the model.

Therefore, we have decided not to include this feature in the model. However, we have extended the discussion in the supplementary text to reflect this point.

*General*

*This study is elegant and the use of dynamical measurements and modeling is synergistic. Using negative feedback (and comparing it to integral feedback) and its features has been heavily studied (see for example, Khammash M. BMC Biology 2016).*

We thank the reviewer for this comment, and we draw your attention to the fact that this article was already cited in the manuscript.

*Feedback terms without saturation seldom describe the data well in biological systems. The reader would benefit from the use of the linear model as a pedagogical example to highlight the fact that overshoot ("training") can happen without dependence on initial conditions and hysteresis ("stress tolerance) also if it appears in the Supplementary Information. The paper would benefit from comparing the Michaelian model to a model where the growth rate is not constant in time. This will provide a vivid example for the interplay between growth rate and transnational feedback.*

This comment is linked to the previous one regarding the effect of non-constant growth rate, and was addressed above.

*Reviewer #2:*

[…] The information obtained from the study is mostly not novel, in contrast of what the authors claim. It has been known for oxidative stress and other stresses, in yeast and in other organisms, that 1) different kinetics of stress exposure leads to different adaptation patterns (training), specially on those systems in which transcription is involved.

Except in the paper from the Klipp group mentioned above, to the best of our knowledge, the hypothesis that stress rate determines cellular adaptation has never been formalized and quantitatively measured, as we did. Experimentally, one study in bacteria has characterized a stress-rate dependent signaling in

We would like to emphasize the fact that the “training” effect reported in our manuscript could be misinterpreted as an acquisition of tolerance to stress. We believe that our study cancels the confusion that can be made between the two phenomena, and clearly exposes the mathematical implications of each behavior.

*And 2) That the pre-exposure to stress leads to acquisition of tolerance. It is also known that the Yap1 transcription factor is a major regulator of oxidative stress defense genes and that Yap1 regulates peroxiredoxin expression (major H_2_O_2_ scavengers). Thus, albeit nicely presented and quantitatively assessed, the new knowledge generated in the study is rather limited.*

As acknowledged by the reviewer, our quantitative approach and its outcome is the main scope of the paper. We think that it provides a strong added value to the understanding of stress response dynamics. For instance, it allows to capture the phenomenon of acquisition of tolerance, which has long been described, but the mechanism of which was unknown.

On top of this, several novel model-independent biological insights have been revealed by our study:

1) Ramp experiments allowed us to unambiguously identify amongst all the cellular H_2_O_2_ scavengers, those that are uniquely required for H_2_O_2_ adaptation (peroxiredoxins), which contrast with other such enzymes that may only contribute to growth rate recovery upon acute stress exposure (e.g. Ccp1, see Figure 5). This distinction is important to decipher the seeming redundancy between the yeast genes that can degrade H_2_O_2_.

2) The hormetic effect of external H_2_O_2_ on replicative lifespan is new and will be of great interest to the aging community.

*The experimental set up used to study adaptation is interesting and well executed. They mostly monitor Yap1 localization as a measure of oxidative stress signaling and growth rate to assess adaptation. These measurements are correct and they provide relevant information however, they fall short to conclude on cellular adaptation. Cell survival should be monitored (at least for Figure 1–Figure 4) to be able to claim on adaptation, measuring growth rate to visualize a transient cell cycle arrest is clearly not sufficient.*

We agree with the reviewer that, so far, our assessment of cellular adaptation to stress only used cellular growth rate and cell division as proxy, and not cell survival. In addition, we acknowledge that our description of the cell fate phenotypes following a H_2_O_2_ step lacked clarity and could lead to reader misinterpretation.

Therefore, we have performed a series of experiments to quantify further cell fate following stress exposure (as explained in detail below) and we have revised the corresponding text in the manuscript.

Upon abrupt exposure to intermediate stress levels (0.4-0.5mM), there is cell fate heterogeneity, as monitored by growth rate, cell division rate and other markers (e.g. DNA damage). While all cells experience a transient decrease (or arrest) in growth rate right after stress exposure (Figure 1), some do not recover a normal growth rate over the course of the experiment (up to 1000 minutes), as quantified in Figure 2.

Similarly, above 0.6mM, all cells experience a growth arrest that is irreversible over the course of the experiment (up to 1000 minutes). We have referred to this phenotype as a “metabolic arrest” in the original submission, because of the absence of cell volume increase. To insist on irreversibility, in the revised text, we now refer to it as “permanent growth arrest”.

As far as we understand the main point of reviewer #2, the permanent growth arrest does not necessarily equate to cell death. Therefore, a complementary survival analysis is required to prove that stalled cells will eventually die.

To test this hypothesis, first, we used a vital stain (propidium iodide, PI) to characterize cell survival upon exposure to a critical dose of 0.6mM H_2_O_2_. Under these conditions, we observed that cells ultimately undergo lysis (as monitored by elevated intracellular PI fluorescence level). Interestingly, this process takes a couple of days, probably because the loss of cell membrane integrity occurs much later than cell growth arrest following exposure to H_2_O_2_. These experiments, which are reported in a new supplementary Figure 2—figure supplement 2, provide the demonstration that cells which are permanently arrested ultimately die at 0.6mM H_2_O_2_. Therefore, since all the cells undergo a permanent growth arrest at this concentration, it is very likely that this growth defect phenotype indicates a failure to undergo adaptation.

In addition, we emphasized the data of the original Figure 2—figure supplement 1, (now Figure 2—figure supplement 2) which aimed at determining the reversibility of the permanent growth arrest phenotype at 0.6mM. In these experiments, we asked whether cells exposed to a “lethal” dose of H_2_O_2_ (0.6mM) for a limited duration (up to 4 hours) would recover a normal growth rate. We found that growth rate would not resume if duration of stress exposure lasted more than 4 hours, therefore suggesting that these cells would certainly not adapt in the long term.

To support this conclusion further, using a GFP-promoter fusion, we observed that expression of the Yap1 target Trx2 vanished following stress exposure, and did not recover when the stressor was removed after 4 hours. This result also support the notion that permanently arrested cells are unable to adapt, because *TRX2* is essential for adaptation.

To check the consistency of this observation at different H_2_O_2_ levels, we performed the same analysis (using a Trx2-GFP protein fusion) at 0.4mM H_2_O_2_ on the subpopulation of cells that undergoes a permanent growth arrest. Similarly as at 0.6mM H_2_O_2_ (lethal dose), we observed a decline in Trx2 level in these cells (but not in the rest of the population), see Figure 2—figure supplement 3 in the revised version.

Furthermore, we noticed that permanently arrested cells tend to display much larger vacuoles and become brighter than adapting cells. We have quantified this phenomenon at both 0.4mM and 0.6mM, which confirms that the permanent growth arrest phenotype is consistent at these two concentrations, see Figure 2—figure supplement 3 in the revised version.

Last, we have verified that the growth arrest phenotype described in step experiment is consistently observed in ramp experiments when the ramp slope is higher than 4.4µM/min (Figure 3—figure supplement 2), as well in experiments related to the acquisition of tolerance (Figure 4 and Figure 4—figure supplement 1: the growth arrest is accompanied by the loss of *TRX2* expression for the critical ΔI and by the presence of bright vacuoles).

Altogether, these new experiments indicate that the permanent growth arrest phenotype, corresponds to cells which fail to adapt in the long term and ultimately die.

We have extensively edited the manuscript to include these new control experiments.

*Actually, cells delay cell cycle in response to stress and this delay is essential to maximize cell survival. The delay per se is an adaptive response and thus it cannot be concluded that cells that delay cell cycle are those that do not adapt properly.*

As pointed above, all cells display a transient growth arrest (and hence a transient cell cycle arrest) following an abrupt exposure to stress. However, there is a subpopulation of cells (that we refer to as “prolonged cell cycle arrest”) that, unlike the adapting cells, does not resume proper cell division, even if it recovers a normal growth rate. In the original submission, we further demonstrated that this phenotype was associated with the activation of the DNA damage response.

Even though we cannot exclude that these cells would ultimately repair the damages on timescales longer than the course of the experiment (> 1000min), the fact that they accumulate a long delay compared to the adapting subpopulation means that they will be outgrown by the adapting cells, which recover a normal division rate. Therefore, knowing whether these cells ultimately adapt would be somewhat irrelevant when considering the entire population of cells.

Altogether, our analysis indicates that cell fate upon exposure to abrupt stress can be split into three qualitatively distinct phenotypes (and this is actually another novel aspect of our study compared to the existing literature). It is indeed remarkable that, unlike what is pointed out by the reviewer, the cells that display a prolonged cell cycle arrest do not resume division even after a few hours, even though their growth rate is back to normal.

The revised version of the text clarifies the distinction between the three observed phenotypes.

*Ramping experiments are interesting (Figure 3) to test the initial hypothesis. However, if I understood them properly, they do not reach the same H_2_O_2_ concentration. Authors should perform the same experiments increasing H_2_O_2_ at different rates but reaching the same final concentration (with different times) and then compare adaptation. Otherwise they are comparing not only different rate of exposure but also different final concentration of H_2_O_2_.*

This is a fair concern of the reviewer. Remarkably, however, this control experiment is already implicitly included in the ramp experiments presented in the phase diagram in Figure 3: if we arbitrarily choose a H_2_O_2_ concentration of 4mM (which corresponds to a virtual vertical line on Figure 3 set by I=4mM), we see that cells are well adapted when using a rate δ=4.4µM/min but have already started to decrease their mean growth rate at δ=8.8µM/min. Therefore, we can conclude that cells exposed to the same final concentration (4mM) but with different rates (4.4 vs 8.8 µM/min) display different cell fates.

To make this point clearer, we have measured the fraction of cells that do not have a permanent growth arrest phenotype as a function of the H_2_O_2_ concentration during a ramp at either 4.4 or 8.8 µM/min. This analysis confirms that cells exposed to 8.8 µM/min start to undergo growth arrest and become vacuoled (at around 2.5mM) while no growth defect is observed at 4.4 µM/min in this range of concentrations. This analysis has been included in the revised version of the manuscript in Figure 3—figure supplement 2.

*Negative feedback is proposed to explain some of the results. The authors do not study the contribution of such feedback in the training or acquired tolerance. This could be an interesting aspect to be analyzed.*

We are not sure how to understand this comment, as the nonlinear negative feedback mediated by peroxiredoxins is the central scope of this paper, which investigates how this regulatory motif induces the emergence of non-intuitive phenomena, such as training, acquisition of tolerance and hormesis.